# Deep Models for Ground Reaction Force Prediction: Subject-Held-Out Evaluation Across Locomotion and Occupational Tasks

## Abstract

Deep learning models for predicting ground reaction forces (GRFs) from motion capture markers are typically evaluated within a single laboratory using trial-level splits that permit subject identity leakage. We show that switching from standard trial-level splits to rigorous Leave-One-Subject-Out (LOSO) evaluation reduces $R^2$ by 0.20–0.30 points ($R^2$ 0.86 → 0.66 on locomotion, 0.86 → 0.56 on occupational tasks), providing, to our knowledge, the first side-by-side quantification of this leakage-induced inflation across two distinct task domains under a common evaluation pipeline. Under LOSO, three representative architectures—ImprovedConvFormer (ICF), CNN-LSTM, and CNN—all generalize to unseen subjects with $R^2 > 0.5$, with no single architecture dominating: rankings are task-dependent (ICF best on locomotion, $R^2$=0.655 on GroundLink, 6 subjects; CNN-LSTM best on occupational data, $R^2$=0.606 on Patient Handling, 10 subjects, $p = 0.002$), though between-model differences are modest relative to fold-level variability at these sample sizes. We quantify a consistent hierarchy of feature difficulty across both datasets—vertical forces and CoP components are easiest, mediolateral forces intermediate, and anteroposterior forces consistently hardest—suggesting the ordering reflects how observable each force component is from marker kinematics rather than model-specific limitations.

Attention analysis reveals that ICF learns biomechanically meaningful sparse patterns (97.5% sparsity) organized into three functionally distinct head clusters that align with gait phase structure, with task-dependent differences between locomotion (periodic stride-locked peaks) and occupational tasks (broader, event-driven attention). Combined dual-holdout cross-validation achieves $R^2$=0.587; a matched-subject reanalysis (evaluating single-dataset and combined models on the *same* held-out subjects) shows the apparent Patient Handling gain from pooling is a subject-selection artifact, and we make no directional transfer claim: naïvely pooling the two datasets has a small effect that is not robust across subjects or preprocessing choices. As an exploratory deployment stress test, we also evaluate cross-laboratory transfer: models trained in one laboratory and tested in the other. In this uncontrolled two-laboratory setting, all architectures exhibit severely negative $R^2$ (as low as $-1.5$), despite sharing anatomical markers and output channels, indicating that naïve cross-site deployment is unreliable when tasks, coordinate systems, and calibration protocols differ. We discuss how future work on geometric equivariance and physics-informed constraints could address this limitation, but do not implement such methods here. Code and all experimental outputs are publicly available.

## 1 Introduction

Predicting ground reaction forces (GRFs) from motion capture markers would remove the primary bottleneck in biomechanical analysis: the requirement for instrumented force plates that restrict measurement to controlled laboratory environments. A growing body of work applies deep learning to this task, reporting encouraging accuracy—CNN-LSTM models achieving $R^2$ of 0.74 for vertical GRF prediction during running

(Chen et al., 2025a), and feedforward neural networks reaching correlation coefficients above 0.90 for GRFs from IMU data (Mundt et al., 2020). These results, however, share a methodological pattern that limits their generalization claims: evaluation via *trial-level* random splits within a single laboratory.

When multiple trials from the same subject appear in both training and test sets, models can exploit subject-specific biomechanical signatures—body mass, limb proportions, habitual gait patterns—rather than learning the underlying kinematic-to-kinetic mapping (Horst et al., 2019; Halilaj et al., 2018). We quantify this inflation directly: switching from trial-level splits to LOSO reduces $R^2$ by 0.20–0.30 points on both datasets ($R^2$ from ∼0.86 to ∼0.56–0.66), confirming that a substantial portion of reported accuracy in the literature reflects subject memorization rather than biomechanical generalization. More critically, single-lab studies cannot assess whether learned representations transfer across laboratories with different equipment, coordinate conventions, and task demands.

**Problem setup.** Concretely, we study a sequence-to-sequence regression problem. The input available at test time is *motion-capture markers only*: a $T \times D$ array of body-relative 3D marker coordinates and derived kinematic features ($D \approx 216$), with force plates assumed unavailable at deployment. The target is the concurrent GRF/CoP signal—10 channels per timestep (bilateral vertical, mediolateral, and anteroposterior force, plus two center-of-pressure coordinates per foot). Crucially, the unit of evaluation is the *subject*, not the trial: models are trained on some subjects and tested on entirely held-out subjects (LOSO), reflecting the real deployment target of a *new person*, and—separately, as a stress test—a *new laboratory*. This subject-level protocol is what distinguishes our evaluation from the trial-level random splits common in prior work.

**Scope and limitations.** We state the main caveats up front. (i) Our cross-laboratory results are *exploratory and confounded*: the two laboratories differ simultaneously in task (locomotion vs. occupational handling), coordinate convention, motion-capture and force-plate hardware, calibration, and subject population, so this single two-lab comparison cannot isolate any one cause (e.g., geometry) from the others. (ii) The datasets are *small*—6 and 10 subjects—so all subject-level comparisons, including the architecture rankings, have limited statistical power and should be read as indicative rather than definitive. (iii) Our contribution is *evaluation methodology and empirical analysis within a common pipeline*; we do not claim a new architecture or state-of-the-art accuracy. We revisit these points in Section 8.

This paper makes three contributions toward understanding multi-laboratory GRF prediction.

**Contribution 1: LOSO benchmarking across two task domains.** We perform rigorous subject-held-out (LOSO) evaluation of three representative deep architectures—CNN, CNN–LSTM, and a CNN–Transformer hybrid (ImprovedConvFormer, ICF)—on two heterogeneous GRF datasets (locomotion vs. occupational patient handling), showing that all achieve $R^2 > 0.5$ on unseen subjects but with strongly task-dependent rankings: ICF leads on GroundLink locomotion data ($R^2$=0.655), while CNN-LSTM leads on Patient Handling occupational data ($R^2$=0.606, $p = 0.002$ vs. ICF; Section 6.1). No single architecture dominates, challenging the assumption that a universal "best model" exists for biomechanical force prediction.

**Contribution 2: Feature-level difficulty and interpretability.** We quantify a consistent hierarchy of feature difficulty across datasets—vertical forces and CoP components are easiest, mediolateral forces intermediate, and anteroposterior forces consistently hardest (Section 6.2)—and use attention analysis to relate model focus to biomechanical events such as stance phase and gait cycles (Section 7). The ICF's attention mechanism discovers biomechanically meaningful sparse patterns (97.5% sparsity) organized into three functionally distinct head clusters, with task-dependent differences reflecting the periodic structure of locomotion versus the aperiodic nature of occupational tasks.

**Contribution 3: Exploratory cross-laboratory stress test.** As an exploratory deployment stress test, we evaluate cross-laboratory transfer between two labs with different tasks, coordinate conventions, and equipment, finding severely degraded performance ($R^2$ as low as $-1.5$; Section 6.3). Combined dual-holdout training achieves pooled $R^2 = 0.587$, but a matched-subject reanalysis finds no robust directional transfer benefit from naïve pooling. We argue these results motivate future work on domain adaptation, invariant

feature representations, and geometric or physics-informed models, but treat cross-laboratory generalization as an open problem rather than a solved one.

## 1.1 Implications

These findings carry consequences that extend beyond biomechanics:

- **For evaluation methodology:** LOSO cross-validation is essential for honest reporting but is not the ceiling—no architecture collapses under LOSO in this study. Cross-laboratory testing, which current benchmarks largely ignore, exposes additional challenges.

- **For architecture design:** Architecture choice is task-dependent rather than universal. ICF's convolutional-Transformer hybrid excels on stereotyped locomotion; CNN-LSTM's sequential memory benefits heterogeneous occupational tasks. Our results suggest that, once models are reasonably tuned, differences between modern architectures are smaller than the performance gaps introduced by changes in task and laboratory, highlighting cross-site generalization as an important open problem.

- **For cross-laboratory generalization:** Our exploratory cross-lab experiments reveal severe degradation when tasks, coordinate conventions, and equipment differ simultaneously. Progress likely requires representations with explicit geometric equivariance—SE(3)-invariant architectures, physics-informed constraints, or canonical pose normalization—which we identify as promising directions for future work (Bronstein et al., 2021).

## 1.2 Paper Organization

Section 2 reviews related work on GRF prediction, evaluation pitfalls, and domain adaptation. Section 3 describes the two datasets and our preprocessing pipeline. Section 4 details the model architectures. Section 5 specifies three evaluation protocols: single-dataset LOSO, combined dual-holdout CV, and cross-lab transfer. Sections 6 and 7 present results and attention analysis. Section 8 discusses cross-laboratory limitations and future directions. Section 9 concludes.

## 2 Related Work

### 2.1 GRF Prediction from Kinematics

Estimating ground reaction forces without force plates has been pursued through both physics-based and data-driven approaches. Physics-based methods use inverse dynamics via musculoskeletal models (Winter, 2009), which can in principle determine GRF from full-body kinematics, but require calibrated segment inertial parameters and complete kinematic chains that are difficult to obtain outside controlled laboratory settings. Data-driven approaches have progressively moved from shallow methods (Oh et al., 2013) to deep learning, with recent work applying feedforward neural networks (Mundt et al., 2020), CNN-LSTM hybrids (Chen et al., 2025a), and other architectures (Stetter et al., 2019) to predict GRFs from marker trajectories or inertial measurements.

Johnson et al. (2021) predicted multidimensional GRFs and moments from wearable sensor accelerations using deep learning, achieving strong performance across multiple movement types. Alcantara et al. (2022) applied bidirectional LSTMs to accelerometer data from 19 runners on varied terrain, reporting RMSE of 0.16 BW for vertical GRF under LOSO evaluation. More recently, Faisal et al. (2024) proposed an attention-guided MultiResUNet that predicts 3D GRFs and moments from foot marker kinematics across 66 subjects from two datasets, reporting cross-dataset Pearson $r = 0.96$—though their evaluation uses trial-level splits rather than subject-held-out protocols. Sugiarto et al. (2025) benchmarked 10 deep learning architectures for GRF prediction during walking, finding that incorporating knee alignment information significantly improves mediolateral force prediction.

Two concurrent works apply learning-based methods to the GroundLink dataset (Han et al., 2023) used in our study. Le et al. (2025) propose a physics-informed approach (Phys-GRD) that enforces Newton–Euler constraints during training on GroundLink; however, they fail to report $R^2$ values directly comparable to our LOSO setting, and use a different evaluation protocol, making direct numerical comparison infeasible. Liu et al. (2025) introduce ImDy, trained on 150+ hours of imitated motion data, and demonstrate zero-shot inverse dynamics estimation that generalizes across datasets including GroundLink—notably outperforming the dataset's own GroundLinkNet baseline without any real GroundLink training data. Both approaches benefit from explicit physical priors or massive pre-training data—directions complementary to the architectural investigation pursued here.

A consistent limitation across this literature is evaluation within a single laboratory using trial-level splits. Chen et al. (2025a) report CNN-LSTM $R^2$=0.74 for vertical GRF prediction during running but evaluate via 10-fold cross-validation without holding out entire subjects. Mundt et al. (2020) predict GRFs from IMU data with cross-validation but do not assess generalization across equipment or task domains. Özates et al. (2025) apply CNNs to cerebral palsy gait data with promising results but evaluate within a single clinical cohort. To our knowledge, no published study has performed LOSO evaluation of GRF prediction simultaneously across two laboratories with differing task types, coordinate systems, and motion capture equipment—the setting we investigate here.

## 2.2 Subject Identity Leakage in Biomechanics

Horst et al. (2019) demonstrated that deep networks can identify individual subjects from a single gait cycle with near-perfect accuracy, implying that subject identity is a strong signal in biomechanical time series. Halilaj et al. (2018) explicitly warned that trial-level splits in biomechanics create "optimistic bias" when subject-specific patterns dominate. Despite these warnings, trial-level evaluation remains the standard practice in most GRF prediction studies.

The leakage problem is not unique to biomechanics. Kaufman et al. (2012) formalized data leakage as information flowing from test labels into the training process, and showed that it is a pervasive source of inflated metrics across applied machine learning. In ecology and environmental modeling, Roberts et al. (2017) established that data with spatial, temporal, or hierarchical structure requires structured cross-validation to avoid inflated metrics—a principle that extends directly to biomechanics, where subjects form a natural hierarchical grouping. Our work applies this same rigor to biomechanical force prediction and quantifies the resulting performance gap.

## 2.3 Cross-Domain Generalization and Geometric Equivariance

Domain adaptation across laboratories or sensor configurations is underexplored in biomechanics. The broader machine learning literature offers extensive work on distribution shift (Quionero-Candela et al., 2009), but most techniques assume shared feature spaces with divergent statistics—a milder condition than the coordinate-system mismatches we encounter.

For biomechanical data specifically, the challenge is geometric: two laboratories measuring the same movement may produce kinematic trajectories that differ by an unknown rotation and translation of the laboratory coordinate frame. This is an instance of the broader problem of learning representations that are equivariant to spatial transformations (Bronstein et al., 2021). Thomas et al. (2018) introduced tensor field networks for SE(3)-equivariant processing of 3D point clouds, and Xu et al. (2024) recently proposed Equivariant Graph Neural Operators (EGNO) for modeling 3D dynamics with provable SE(3) equivariance at ICML 2024. These architectures process geometric data in a frame-independent manner and represent a promising direction for cross-laboratory biomechanical prediction, though neither has been applied to GRF estimation.

Large-scale efforts like AddBiomechanics (Werling et al., 2023) aim to standardize biomechanical data processing across laboratories, but even standardized pipelines cannot resolve fundamental differences in task biomechanics between walking and occupational lifting.

### 2.4 Transformers for Time-Series Regression

The application of Transformers to time-series data has produced mixed results. While specialized architectures like Informer (Zhou et al., 2021) and Autoformer (Wu et al., 2021) achieve strong forecasting performance, Zeng et al. (2023) showed that simple linear models can match or exceed Transformer performance on many benchmarks, questioning whether self-attention provides genuine benefits for temporal data.

For *regression* (predicting concurrent outputs from inputs, not future forecasting), evidence is sparser. Our work contributes to this debate by showing that a carefully designed convolutional Transformer hybrid—with shallow depth, residual CNN features, and Pre-Layer Normalization (Xiong et al., 2020)—can achieve robust generalization where pure recurrent and convolutional architectures fail, at least within a single laboratory domain. One interpretation is that any Transformer advantage under LOSO comes not from capacity but from inductive bias—global self-attention may be less prone than local convolutions or recurrent states to overfitting subject-specific temporal patterns. We present this as an empirical observation rather than a demonstrated mechanism, particularly as the ranking of the three architectures is task-dependent under matched tuning budgets (Section 6.2).

## 3 Datasets and Preprocessing

We evaluate on two independent biomechanics datasets collected in different laboratories with different motion capture systems, coordinate conventions, task types, and subject populations. This deliberate heterogeneity is central to our study: it exposes generalization failures that single-lab evaluations cannot reveal.

**Dataset selection.** These are the two datasets available to us that span genuinely different task domains, rather than a controlled cross-laboratory design. GroundLink was selected as the most complete publicly available marker and force-plate *locomotion* dataset we could identify; the Patient Handling data was contributed by a partner laboratory and provides an *occupational-task* domain with independent acquisition. We emphasize that two datasets differing in task, hardware, coordinate convention, and calibration simultaneously cannot isolate the general cross-laboratory problem; accordingly we treat the cross-lab evaluation (Protocol C) as an exploratory stress test rather than a controlled study (see the Scope and Limitations paragraph in Section 1).

### 3.1 Dataset Descriptions

#### 3.1.1 GroundLink (GL)

The GroundLink dataset (Han et al., 2023) contains motion capture recordings of locomotion tasks (walking, running, jumping, and sport-specific movements) from 7 subjects. Data were collected using a Qualisys Track Manager (QTM) system with dual AMTI force plates. Marker positions were recorded at 250 Hz (downsampled to 50 Hz) and forces at 250 Hz (downsampled to 50 Hz). After excluding subject 1, 6 subjects contribute 283 usable trials (26–57 per subject, unbalanced). Subject 1 was excluded because its anteroposterior force channels are sign-inverted relative to all other subjects and its center-of-pressure offsets exceed 500 mm—values physically implausible for the recorded force-plate geometry and consistent with a data collection or export artifact (e.g., swapped force-plate axes or incorrect plate origin definition) rather than legitimate biomechanical variation. Including subject 1 with a sign correction was considered but rejected because we cannot verify which axis was affected without access to the original calibration files.

GroundLink uses a Z-up coordinate convention: $F_x$ = anteroposterior (AP), $F_y$ = mediolateral (ML), $F_z$ = vertical.

#### 3.1.2 Patient Handling (PH)

The Patient Handling dataset (Chen et al., 2025b) contains recordings of occupational patient handling tasks (lifting, transferring, and repositioning movements) from 12 subjects. Data were collected using a Vicon system with dual AMTI force plates. Marker positions were recorded at 50 Hz and forces at 1000 Hz

(downsampled to 50 Hz). Marker positions and forces share a Y-up OpenSim coordinate convention. After excluding 2 subjects for data quality issues (missing markers, anomalous force profiles), 10 subjects contribute 460 usable trials (46 per subject, balanced).

PH uses a Y-up convention: $v_x = $ ML, $v_y = $ vertical, $v_z = $ AP.

### 3.1.3 Summary

Table 1: Dataset characteristics. Both datasets share 20 anatomical markers and produce 10 output channels (bilateral force + CoP), but differ in every other respect.

| Property | GroundLink | Patient Handling |
|---|---|---|
| Task type | Locomotion | Occupational |
| Subjects (usable) | 6 | 10 |
| Trials (usable) | 283 | 460 |
| Trials/subject | 26–57 (unbalanced) | 46 (balanced) |
| MoCap system | Qualisys (C3D) | Vicon (TRC/MOT) |
| Coordinate frame | Z-up | Y-up (OpenSim) |
| Marker rate | 250 Hz $\rightarrow$ 50 Hz | 50 Hz |
| Force rate | 250 Hz $\rightarrow$ 50 Hz | 1000 Hz $\rightarrow$ 50 Hz |
| Vert. force mean | 249.5 N | 386.2 N |
| AP force std | 33.6 N | 44.3 N |

### 3.2 Common Marker Set

The two datasets use different marker protocols: GL places 24 markers, PH places 20. We identify a 20-marker intersection set spanning bilateral shoulder, elbow, wrist, knee, ankle, heel, toe, pelvis, thigh, and shank landmarks. All models receive input derived exclusively from this common set, ensuring that cross-lab comparisons use identical anatomical features.

### 3.3 Preprocessing Pipeline

All trials undergo a five-stage preprocessing pipeline, applied identically regardless of source laboratory:

**Stage 1: Pelvis root subtraction.** All marker positions are re-expressed relative to the pelvis centroid (mean of left and right ASIS markers), removing absolute spatial position and reducing the feature space to body-relative coordinates.

**Stage 2: Heading alignment.** Each trial is rotated about the vertical axis so that the subject"s forward (anterior) direction aligns with the positive Z-axis. Heading is estimated from the subject"s pelvis anterior–posterior orientation—the vector from the posterior-superior-iliac-spine (PSIS) midpoint to the anterior-superior-iliac-spine (ASIS) midpoint, averaged over the trial—with fallbacks to the ASIS line (LASI–RASI) or the shoulder line when pelvis markers are incomplete. Because forward is fixed by pelvic anatomy, this definition is free of the sign ambiguity that a displacement-based (e.g., principal-component) estimate would introduce. This removes arbitrary laboratory orientation but does not eliminate all cross-lab geometric differences (see Section 6.3).

**Stage 3: Height scaling.** Marker positions are divided by subject height (meters), normalizing for stature differences.

**Stage 4: Feature engineering.** From 20 markers × 3 axes = 60 raw position features, we compute first-order velocity, second-order acceleration, and 10 bilateral relative marker distances, yielding approximately 216 input features per timestep.

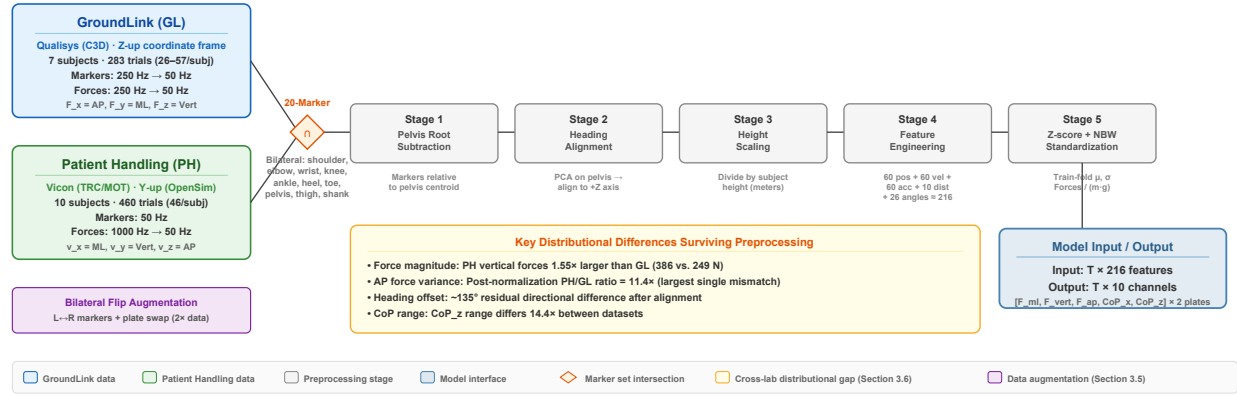

Figure 1: Data preprocessing pipeline. Two independent datasets (GroundLink and Patient Handling) with different motion capture systems, coordinate conventions, and sampling rates are unified through a 20-marker intersection set and a five-stage preprocessing pipeline. Despite preprocessing, substantial distributional differences persist between datasets (yellow box), motivating the diagnostic analysis in Section 3.6.

**Stage 5: Standardization.** Input features are z-score normalized (zero mean, unit variance) with statistics computed only on training fold data. Force targets are normalized by body weight ($F_{\text{NBW}} = F/(m \cdot g)$), then z-score normalized per fold. NBW normalization is critical: without it, models learn body-mass-dependent scaling rather than biomechanical patterns.

Figure 1 illustrates the complete data pipeline from raw laboratory recordings to model-ready tensors, including the key distributional differences that survive preprocessing.

### 3.4 Target Channels

All models predict 10 output channels: bilateral (plate 1 and plate 2) mediolateral force, vertical force, anteroposterior force, and center-of-pressure $x$ and $z$ coordinates. Force channel naming is unified across datasets after coordinate correction:

$$\mathbf{y}_t = [\text{F1}_{\text{ml}}, \text{F1}_{\text{vert}}, \text{F1}_{\text{ap}}, \text{F1}_{px}, \text{F1}_{pz}, \text{F2}_{\text{ml}}, \text{F2}_{\text{vert}}, \text{F2}_{\text{ap}}, \text{F2}_{px}, \text{F2}_{pz}] \tag{1}$$

### 3.5 Data Augmentation

We apply bilateral flip augmentation during training: left-right marker positions and corresponding force plate assignments are swapped, effectively doubling the training set. This leverages the bilateral symmetry of human biomechanics without introducing geometric artifacts.

### 3.6 Key Distributional Differences

Despite preprocessing, substantial distributional differences persist between the two datasets:

- **Force magnitude:** PH vertical forces are $1.55\times$ larger than GL (mean 386 vs. 249 N), reflecting the greater exertion in patient handling tasks.

- **AP force variance:** The post-normalization variance ratio for anteroposterior forces is $11.4\times$ (PH/GL), the single largest distributional mismatch.

- **Heading misalignment:** After preprocessing, the aligned forward direction differs by approximately 135° between datasets—GL subjects' motion lies along $-135°$, PH along 0° in the aligned frame.

- **CoP range:** Center-of-pressure $z$-coordinate ranges differ by $14.4\times$, reflecting different force plate placement conventions.

These differences are not artifacts of our pipeline; they reflect genuine physical and methodological disparities between laboratories. They motivate our diagnostic analysis in Section 6.3.

# 4 Methods

We compare three architectures of increasing representational capacity. Each receives the same $T \times D$ input tensor (a $T$-timestep trial with $D \approx 216$ features) and produces a $T \times 10$ output tensor of bilateral GRF/CoP predictions.

## 4.1 CNN Baseline

Our simplest baseline is a three-layer 1D convolutional network. Each layer applies temporal convolution with kernel sizes 3, 5, and 7 (progressively increasing receptive field), followed by batch normalization and GELU activation (Hendrycks and Gimpel, 2016). Residual skip connections link successive layers. A final linear projection maps from the convolutional feature dimension to 10 output channels. This architecture has approximately 1.2M parameters.

## 4.2 CNN-LSTM Baseline

The CNN-LSTM mirrors the convolutional encoder described above, but replaces the final projection with a two-layer bidirectional LSTM (Hochreiter and Schmidhuber, 1997; Schuster and Paliwal, 1997). The LSTM receives the CNN"s output features at each timestep and produces a hidden state that is linearly projected to 10 outputs. This architecture has approximately 3.1M parameters and is representative of the hybrid CNN-recurrent designs popular in biomechanical prediction (Chen et al., 2025a).

## 4.3 ImprovedConvFormer (ICF)

The ImprovedConvFormer is a hybrid CNN-Transformer architecture designed to combine local feature extraction with global temporal attention.

### 4.3.1 Convolutional Encoder

The encoder consists of three residual convolutional blocks. Each block applies a 1D convolution with kernel sizes $k \in \{3, 5, 7\}$, followed by batch normalization and GELU activation. A residual connection bypasses each block via a $1 \times 1$ convolution when the channel dimension changes. The encoder maps the input from $D$ features to $d_{\mathrm{model}}$ channels:

$$\mathbf{H}_{\mathrm{cnn}} = \mathrm{ConvEncoder}(\mathbf{X}) \in \mathbb{R}^{T \times d_{\mathrm{model}}} \tag{2}$$

### 4.3.2 Metadata Integration

Optional scalar metadata (subject attributes—in our experiments, mass and height) is projected to $d_{\mathrm{model}}$ via a linear layer and added to every timestep by broadcast addition:

$$\mathbf{H}_{\mathrm{meta}} = \mathbf{H}_{\mathrm{cnn}} + \mathrm{Linear}(\mathbf{m}) \tag{3}$$

where $\mathbf{m}$ is the metadata vector. In all reported experiments the metadata vector consists solely of subject mass and height; no dataset or laboratory indicator is provided to the model.

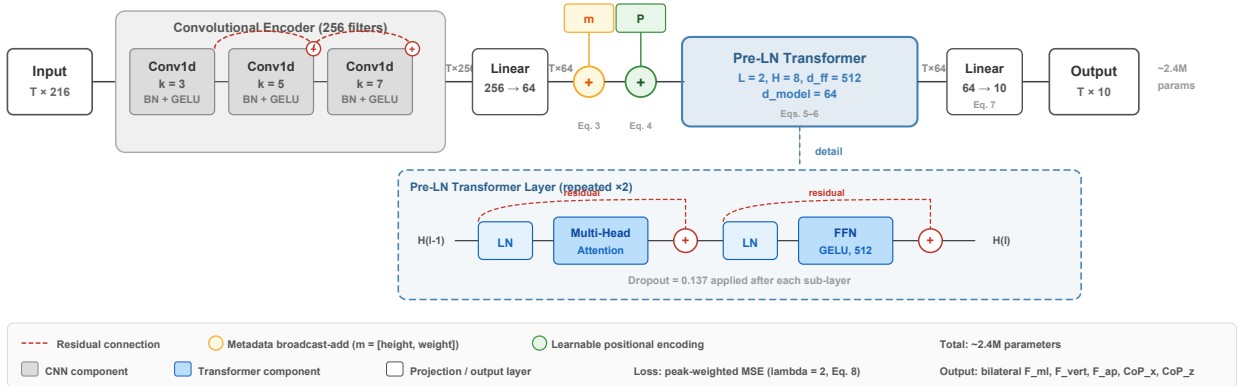

Figure 2: ImprovedConvFormer (ICF) architecture. An input sequence ($T \times D$, $D \approx 216$ features) passes through a three-block residual CNN encoder (kernel sizes 3, 5, 7; 256 filters) with batch normalization and GELU activation. A linear projection reduces dimensionality to $d_{\mathrm{model}} = 64$, after which metadata (height, weight) is broadcast-added and learnable positional encodings are applied. A 2-layer Pre-LN Transformer with 8 attention heads processes the enriched sequence, and a linear output head produces 10 bilateral GRF/CoP predictions at every timestep. Training uses a peak-weighted loss ($\lambda = 2$) that emphasizes stance-phase force peaks. Total: ~2.4M parameters.

### 4.3.3 Positional Encoding

A learnable positional embedding $\mathbf{P} \in \mathbb{R}^{T_{\max} \times d_{\mathrm{model}}}$ is added to the convolutional features. Unlike sinusoidal encodings, learnable embeddings allow the model to discover task-specific temporal structure:

$$\mathbf{H}_{\mathrm{pos}} = \mathbf{H}_{\mathrm{meta}} + \mathbf{P}_{:T} \tag{4}$$

### 4.3.4 Pre-LN Transformer Layers

The Transformer component uses the Pre-LayerNorm variant (Xiong et al., 2020), which applies layer normalization *before* rather than after each sub-layer. This ordering stabilizes training at deeper scales. Each of $L$ identical layers applies:

$$\hat{\mathbf{H}}^{(\ell)} = \mathbf{H}^{(\ell-1)} + \mathrm{MHA}\big(\mathrm{LN}(\mathbf{H}^{(\ell-1)})\big) \tag{5}$$

$$\mathbf{H}^{(\ell)} = \hat{\mathbf{H}}^{(\ell)} + \mathrm{FFN}\big(\mathrm{LN}(\hat{\mathbf{H}}^{(\ell)})\big) \tag{6}$$

where MHA is multi-head self-attention with $n_{\mathrm{head}}$ heads, FFN is a two-layer feedforward network with GELU activation, and LN is layer normalization.

### 4.3.5 Output Head

The Transformer"s output is projected to 10 channels by a final linear layer applied at every timestep:

$$\hat{\mathbf{y}}_t = \mathbf{W}_{\mathrm{out}}\mathbf{H}_t^{(L)} + \mathbf{b}_{\mathrm{out}} \in \mathbb{R}^{10} \tag{7}$$

### 4.3.6 Architecture Summary

Figure 2 illustrates the end-to-end ICF pipeline. Table 2 summarizes the default hyperparameters used across all experiments unless otherwise stated.

Table 2: ICF architecture hyperparameters. Wherever applicable, these were tuned via Optuna-based HPO (Section 5.2).

| Hyperparameter | Value |
|---|---|
| Conv. filters ($n_{\text{filters}}$) | 256 |
| Conv. kernel sizes | 3, 5, 7 |
| Transformer layers ($L$) | 2 |
| Attention heads ($n_{\text{head}}$) | 8 |
| Model dimension ($d_{\text{model}}$) | 64 |
| Feedforward dimension | 512 |
| Dropout | 0.137 |
| Learnable positional encoding | Yes |
| Total parameters | $\approx$2.4M |

### 4.4 Loss Function

All models are trained with a *normalized peak-weighted* loss that up-weights high-magnitude regions of each GRF/CoP channel. Within each mini-batch, every output channel is min–max normalized to $[0, 1]$, and a per-sample weight $w_{t,c} = 1 + \lambda \cdot \tilde{y}_{t,c}$ is applied, where $\tilde{y}_{t,c}$ is the normalized target and $\lambda = 2$ is the peak-weight multiplier:

$$\mathcal{L} = \frac{1}{T \cdot C} \sum_{t=1}^{T} \sum_{c=1}^{C} w_{t,c} \left( \hat{y}_{t,c} - y_{t,c} \right)^2, \qquad w_{t,c} = 1 + \lambda \, \tilde{y}_{t,c} \tag{8}$$

where $C = 10$ is the number of output channels. The peak weighting ensures that stance-phase peaks (which carry the most biomechanical information) contribute up to $3\times$ more to the gradient than swing-phase near-zero values. We evaluate model performance using the coefficient of determination ($R^2$) computed per channel and averaged:

$$R_c^2 = 1 - \frac{\sum_t (\hat{y}_{t,c} - y_{t,c})^2}{\sum_t (y_{t,c} - \bar{y}_c)^2}, \qquad \bar{R}^2 = \frac{1}{C} \sum_{c=1}^{C} R_c^2 \tag{9}$$

Negative $R^2$ values indicate predictions worse than predicting the test-set mean, a condition we frequently observe in cross-lab transfer experiments.

## 5 Experimental Setup

### 5.1 Evaluation Protocols

We design three nested evaluation protocols of increasing stringency. Each protocol answers a distinct scientific question.

#### 5.1.1 Protocol A: Single-Dataset Leave-One-Subject-Out (LOSO)

For each dataset independently, we perform leave-one-subject-out cross-validation. In each fold, all trials from one subject serve as the test set; the remaining subjects form the training set. This yields 6 folds for GL and 10 folds for PH. LOSO eliminates subject identity leakage by construction: the test subject"s movement patterns are never seen during training.

This protocol is our primary diagnostic tool. By comparing LOSO performance to any trial-split baseline, we quantify the magnitude of leakage inflation.

### 5.1.2   Protocol B: Combined Dual-Holdout Cross-Validation

To evaluate multi-lab training, we combine both datasets and perform a 6-fold cross-validation where each fold holds out one GL subject *and* one PH subject simultaneously. The subject pairings are:

Table 3: Combined dual-holdout CV fold assignments.

| Fold | GL held-out | PH held-out |
|:---:|:---:|:---:|
| 1 | gl_2 | ph_1 |
| 2 | gl_3 | ph_2 |
| 3 | gl_4 | ph_3 |
| 4 | gl_5 | ph_4 |
| 5 | gl_6 | ph_5 |
| 6 | gl_7 | ph_6 |

At test time, each fold"s held-out GL subject is evaluated only on GL test trials and each held-out PH subject only on PH test trials. This protocol trains a single model on both labs" data and tests whether multi-lab pooling improves or degrades within-domain accuracy.

### 5.1.3   Protocol C: Cross-Lab Transfer

The most stringent evaluation trains on one dataset entirely and tests on subjects from the other. Specifically, in Protocol A"s LOSO folds, we also record the held-out model"s predictions on all subjects from the *other* dataset. This yields a cross-lab $R^2$ for each fold, measuring pure domain transfer without any target-domain training data.

### 5.2   Hyperparameter Optimization

We use Optuna (Akiba et al., 2019) with a Tree-structured Parzen Estimator (TPE) sampler to tune hyperparameters. The search space includes:

Table 4: Hyperparameter search space and best configuration (combined CV).

| Hyperparameter | Search Range | Best |
|:---|:---:|:---:|
| Learning rate | [1e-4, 1e-2] (log) | 4.6e-4 |
| Dropout | [0.05, 0.3] | 0.137 |
| $d_{\mathrm{model}}$ | {32, 64, 128, 256} | 64 |
| Batch size | {8, 16, 32, 64} | 16 |
| Transformer layers | {1, 2, 3, 4} | 2 |
| Attention heads | {4, 8} | 8 |
| Conv. filters | {64, 128, 256} | 256 |
| Feedforward dim. | {128, 256, 512} | 512 |
| Weight decay | [5e-3, 1e-1] (log) | 7.36e-4 |

HPO is run for 100 trials per configuration. To prevent information leakage during HPO, we use an inner LOSO loop: the objective function is the mean validation $R^2$ across all training-set subjects, with no access to the held-out test subject.

### 5.3   Training Details

All models are trained with the AdamW optimizer (Loshchilov and Hutter, 2019) for up to 200 epochs with early stopping (patience = 15 epochs, monitored on validation loss). The learning rate follows a cosine annealing schedule with warm restarts. Gradient clipping is applied at a maximum norm of 1.0.

Training and evaluation are performed on a single NVIDIA RTX 3090 GPU. A complete 6-fold combined CV run (including HPO) requires approximately 8 hours for ICF.

### 5.4 Statistical Reporting

Unless otherwise noted, we report mean ± standard deviation across LOSO folds. For significance testing between models, we use the Wilcoxon signed-rank test across fold-level $R^2$ values with $\alpha = 0.05$, which is appropriate for the small, paired, non-normally-distributed samples in our LOSO folds ($n = 6$ for GL, $n = 10$ for PH). All per-fold results are tabulated in Appendix A.1.

## 6 Results

We organize results around three findings: (1) within-domain LOSO performance reveals task-dependent architecture rankings, (2) combined multi-lab training yields no robust directional transfer once held-out subjects are matched, alongside a consistent feature difficulty hierarchy, and (3) an exploratory cross-laboratory stress test reveals severe transfer limitations.

### 6.1 Finding 1: LOSO Reveals Task-Dependent Architecture Rankings

Table 5 summarizes LOSO results across all model–dataset combinations. The headline result is that *no single architecture dominates*: the best model depends on the dataset.

Table 5: Single-dataset LOSO $R^2$ and MAE (mean across folds). ICF leads on GL; CNN-LSTM leads on PH. Wilcoxon signed-rank tests are reported in the text.

| Model | Dataset | LOSO $R^2$ | MAE | Folds |
|---|---|---|---|---|
| ICF | GL | 0.655 | 0.328 | 6 |
| ICF | PH | 0.561 | 0.400 | 10 |
| CNN-LSTM | GL | 0.619 | 0.351 | 6 |
| CNN-LSTM | PH | 0.606 | 0.374 | 10 |
| CNN | GL | 0.530 | 0.405 | 6 |
| CNN | PH | 0.555 | 0.413 | 10 |

On GroundLink (locomotion), the architecture ranking is ICF > CNN-LSTM > CNN. ICF significantly outperforms CNN ($W = 0$, $p = 0.031$) but not CNN-LSTM ($W = 3$, $p = 0.156$).

On Patient Handling (occupational tasks), the ranking *reverses* for the top two models: CNN-LSTM ($R^2 = 0.606$) significantly outperforms ICF ($R^2 = 0.561$, $W = 0$, $p = 0.002$). CNN ($R^2 = 0.555$) is not significantly different from ICF ($W = 21$, $p = 0.557$).

We note that statistical power is limited at these sample sizes: with $n = 6$ (GL) and $n = 10$ (PH) folds, the minimum achievable $p$-value under the Wilcoxon signed-rank test is 0.031 and ~0.002, respectively. The observed $p$-values for significant comparisons are therefore near the floor of what the test can detect, and should be interpreted with caution.

**Non-deep baseline.** To separate the contribution of temporal modeling from that of the preprocessing pipeline and feature set, we add a per-channel ridge-regression baseline (RidgeCV; regularization chosen by generalized cross-validation on training frames only) evaluated under *identical* LOSO folds, features (216-dim), within-fold normalization, and per-channel $R^2$ metric. This model is deliberately frame-wise and memoryless, so it isolates what a linear map from marker kinematics to force can achieve without temporal context.

On identical features the linear model is at chance on both datasets; even vertical force—easiest for the deep models—is near zero, because instantaneous force is not recoverable from a memoryless map of marker positions. The deep models' accuracy therefore reflects temporal modeling, not the feature pipeline.

Table 6: Non-deep (ridge) LOSO baseline vs. the deep models, under identical folds, features, and normalization. A memoryless linear map from marker kinematics to force is at chance on both datasets, so the deep models' accuracy reflects temporal modeling rather than the preprocessing pipeline.

| Model | GL LOSO $R^2$ | PH LOSO $R^2$ |
|---|---|---|
| Ridge (linear, memoryless) | $0.05 \pm 0.05$ | $-0.02 \pm 0.09$ |
| CNN | 0.530 | 0.555 |
| CNN-LSTM | 0.619 | 0.606 |
| ICF | 0.655 | 0.561 |

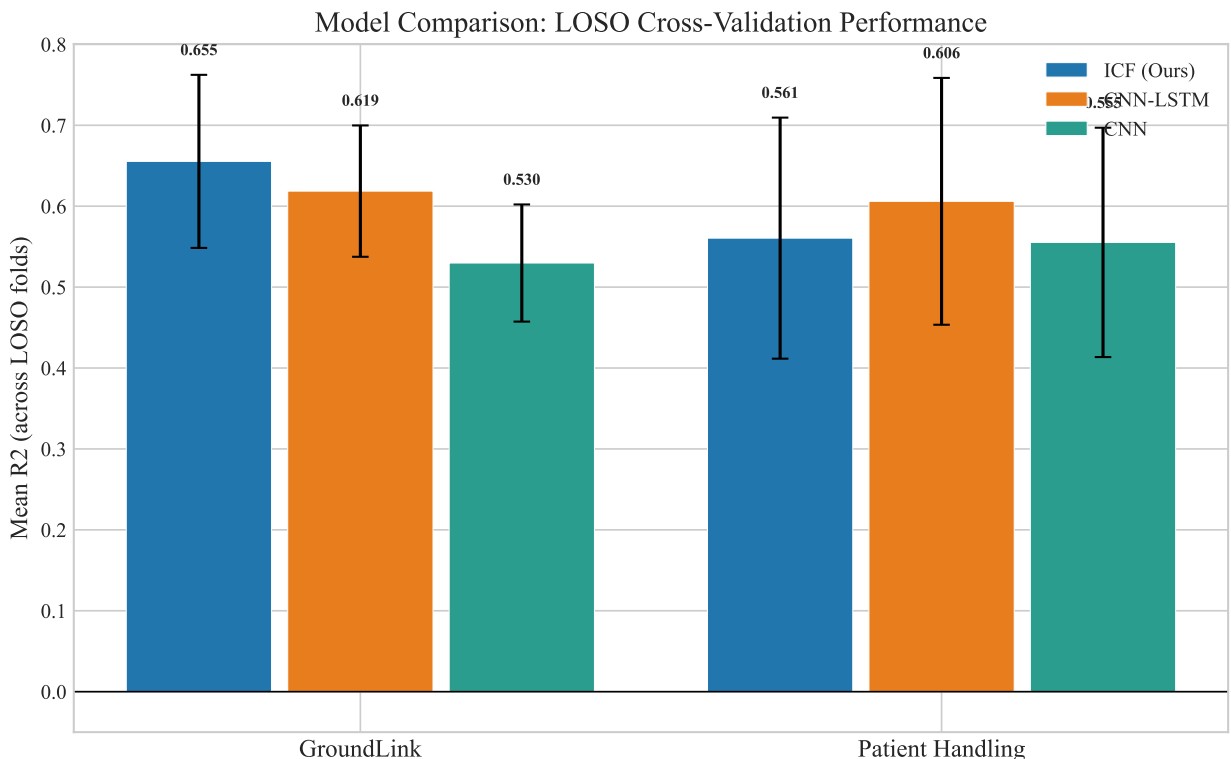

Figure 3: LOSO $R^2$ for each model–dataset combination. Architecture ranking reverses between datasets: ICF leads on GL while CNN-LSTM leads on PH. All models maintain $R^2 > 0.5$ on both datasets.

### 6.1.1 Comparison with Published Methods

Table 7 places our LOSO results in context with recent GRF prediction studies. Direct numerical comparison across rows is constrained by differences in tasks, input modalities, evaluation protocols, output channels, and reported metrics. Several important caveats apply:

- $r$ **vs.** $R^2$: Pearson correlation ($r$) measures linear association; $R^2$ (coefficient of determination) measures variance explained. For single-output univariate predictions $r^2 = R^2$, but for multi-output models with per-channel aggregation—as in our 10-channel setting—$R^2$ can be substantially lower than $r^2$ because it penalizes bias and scale errors that $r$ ignores. Studies reporting $r$ are therefore not directly comparable with those reporting $R^2$.

- **MSE and mPJE:** Mean squared error (MSE) and mean per-joint error (mPJE, in N/kg) depend on target scaling and cannot be converted to $R^2$ without access to target variance. mPJE specifically measures per-joint force error normalized by body mass, as defined by Liu et al. (2025).

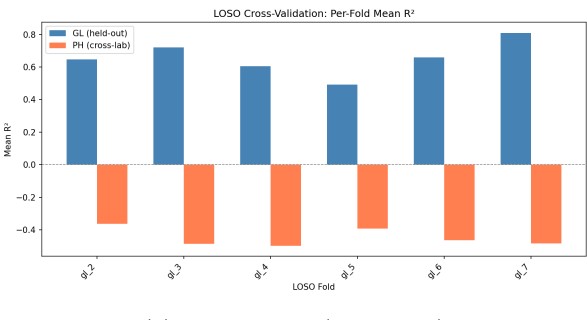 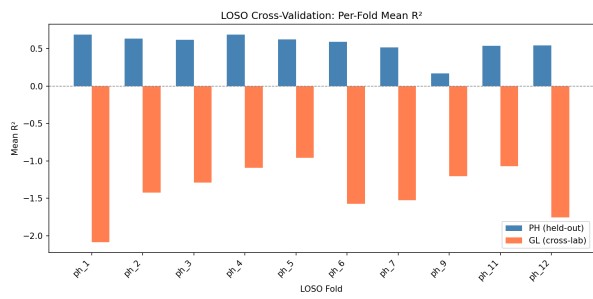

(a) GroundLink (6 subjects)   (b) Patient Handling (10 subjects)

Figure 4: ICF per-subject LOSO $R^2$. GL shows tight consistency ($R^2$ range: 0.49–0.81); PH reveals greater inter-subject variability, with subject ph_9 as a clear outlier ($R^2 = 0.17$). Per-channel inspection reveals that ph_9's low score is driven by catastrophic failure on foot 2 force channels ($R^2 < 0$ for ML, vertical, and AP forces on plate 2), while foot 1 channels remain reasonable ($R^2 = 0.40$–$0.89$). This bilateral asymmetry suggests a subject-specific biomechanical pattern or data quality issue in the second force plate that the model cannot generalize to from the remaining 9 subjects.

Table 7: Comparison with recent GRF prediction methods. **Eval.**: LOSO = leave-one-subject-out, Trial = trial-level split, LOOCV = leave-one-out CV. **Out.**: F = forces only, F+CoP = forces + centers of pressure, F+M = forces + moments. **Metric**: the primary accuracy metric reported by each study in its native units; metrics are *not* comparable across rows due to differences in normalization, output channels, and evaluation protocols. [†]Trained on simulated data only (zero-shot transfer to GroundLink subject 7). [‡]MSE$\times10^{-3}$, body-weight-normalized. Our results (shaded) are the only multi-laboratory LOSO evaluations predicting both forces and CoP.

| Study | Task | $N_{\mathbf{subj}}$ | Eval. | Outputs | Reported Metric |
|---|---|---|---|---|---|
| Johnson et al. (2021) | Athletic | 15 | LOOCV | F+M | (deep learning, IMU) |
| Alcantara et al. (2022) | Running | 19 | LOSO | F | RMSE 0.16 BW ($F_z$) |
| Faisal et al. (2024) | Walking | 66 | Trial | F+M | $r=0.96$ (cross-dataset) |
| Chen et al. (2025a) | Running | 12 | 10-fold | F | $R^2=0.74$ ($F_z$, CNN-LSTM) |
| Sugiarto et al. (2025) | Walking | — | 5-fold | F | (architecture survey) |
| Han et al. (2023) | Locomotion | 7 | LOSO | F+CoP | MSE 0.18[‡] |
| Liu et al. (2025)[†] | Mixed | 0 | Zero-shot | F | mPJE 1.07 N/kg |
| Le et al. (2025) | Locomotion | — | — | F | (no $R^2$ reported) |
| ICF (Ours) | Locomotion | 6 | LOSO | F+CoP | $R^2=0.655$, MAE=0.328 |
| CNN-LSTM (Ours) | Occupational | 10 | LOSO | F+CoP | $R^2=0.606$, MAE=0.374 |

- **Output scope:** Studies predicting only vertical force ($F_z$) or only 3D forces (F) solve a simpler problem than predicting forces and centers of pressure (F+CoP), which adds four CoP channels with different units and variability.

To facilitate the most informative comparison possible, we report both $R^2$ and MAE for our own results.

**Leakage inflation.** To quantify the impact of subject leakage, we compare LOSO results against standard trial-level random splits (80/10/10) using the same ICF architecture. Trial-split evaluation yields $R^2 = 0.860$ on GL and $R^2 = 0.856$ on PH—inflated by +0.205 and +0.295 respectively relative to LOSO. This 24–35% inflation confirms that a substantial portion of accuracy reported under trial-level protocols reflects subject memorization rather than generalizable biomechanical learning.

**Interpretation.** The task-dependent ranking reflects architectural biases. CNN-LSTM's bidirectional recurrence is well-suited to the variable-length, aperiodic movements in patient handling, where temporal

context spanning seconds can inform force predictions. ICF's attention mechanism, by contrast, excels at capturing the stereotyped, periodic gait cycles in locomotion data, where local convolutional features align well with stride-phase structure. Notably, all three architectures generalize meaningfully to unseen subjects under LOSO—none collapses, and all produce $R^2 > 0.5$ on both datasets.

### 6.2 Finding 2: Within-Domain Performance and Combined Training

#### 6.2.1 Feature-Level Analysis

Not all GRF/CoP channels are equally predictable. Table 8 reports per-channel $R^2$ for the ICF under single-dataset LOSO:

Table 8: Per-channel ICF LOSO $R^2$ for GL and PH. Vertical force and CoP position are well-predicted; AP force remains challenging.

| Channel | GL $R^2$ | PH $R^2$ |
|---|---|---|
| Vertical force (F1) | 0.802 | 0.708 |
| Vertical force (F2) | 0.761 | 0.539 |
| ML force (F1) | 0.690 | 0.517 |
| ML force (F2) | 0.720 | 0.371 |
| AP force (F1) | 0.410 | 0.544 |
| AP force (F2) | 0.311 | 0.148 |
| CoP $x$ (F1) | 0.862 | 0.847 |
| CoP $z$ (F1) | 0.515 | 0.658 |
| CoP $x$ (F2) | 0.786 | 0.725 |
| CoP $z$ (F2) | 0.697 | 0.547 |
| **Mean** | **0.655** | **0.561** |

A clear hierarchy emerges:

1. **Tier 1** ($R^2 > 0.7$): Vertical force and CoP $x$—these channels are tightly coupled to stance phase kinematics (heel-strike, mid-stance, toe-off) and are predictable from lower-limb marker trajectories.

2. **Tier 2** ($0.5 < R^2 < 0.7$): ML force and CoP $z$—predictable but with greater inter-subject variability, reflecting individual lateral balance strategies.

3. **Tier 3** ($R^2 < 0.5$): AP force—consistently challenging. AP forces depend on instantaneous acceleration and braking, which are minimally observable from position-derived features alone.

Figure 5 visualizes this hierarchy, and Figure 6 further illustrates the challenge of AP force prediction across folds, showing that this channel exhibits the highest inter-fold variability and the largest domain gap between GL and PH. The Tier 1/2/3 ranking is consistent across all three architectures and both datasets (per-architecture breakdowns are provided in Appendix A.6).

#### 6.2.2 Combined Dual-Holdout Cross-Validation

Under Protocol B (combined CV), the ICF trained on both datasets achieves a pooled $R^2 = 0.587$ across 6 folds (Table 9).

At face value, PH performance appears to *improve* from $R^2 = 0.561$ (single-dataset LOSO) to $R^2 = 0.610$ (combined CV), while GL degrades from $R^2 = 0.655$ to $R^2 = 0.568$. These comparisons are, however, confounded by subject selection: the combined folds hold out only six of the ten PH subjects, and those six are easier than average—they reach $R^2 = 0.639$ under PH-only LOSO, well above the 10-subject mean of 0.561. When the comparison is repeated on *matched* held-out subjects (single-dataset LOSO restricted to the same subjects used in the combined folds), the apparent PH gain disappears (Table 10), and the

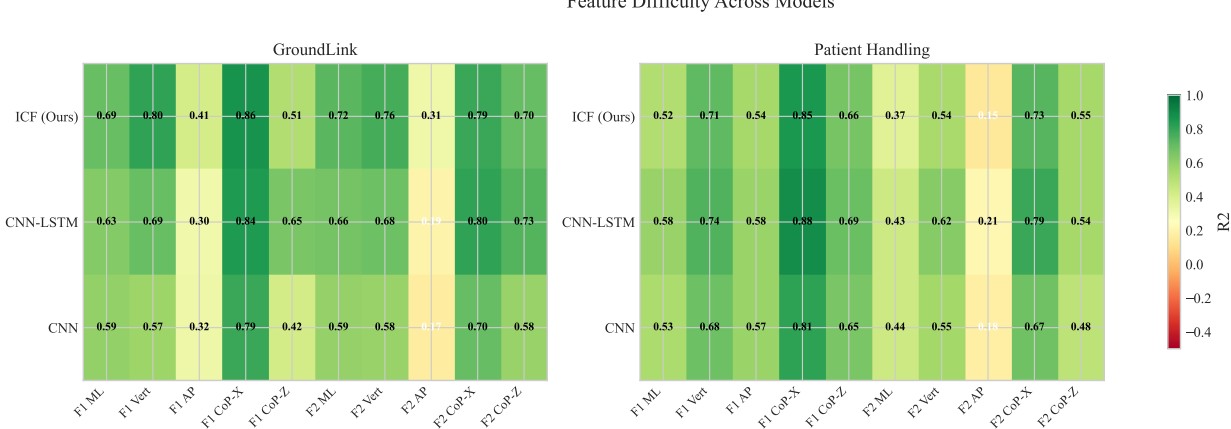

Figure 5: Feature difficulty ranking. Per-channel $R^2$ (mean across models and datasets) sorted by difficulty. The gap between Tier 1 channels ($R^2 > 0.7$) and Tier 3 AP force ($R^2 < 0.4$) reflects differences in how observable each force component is from marker positions.

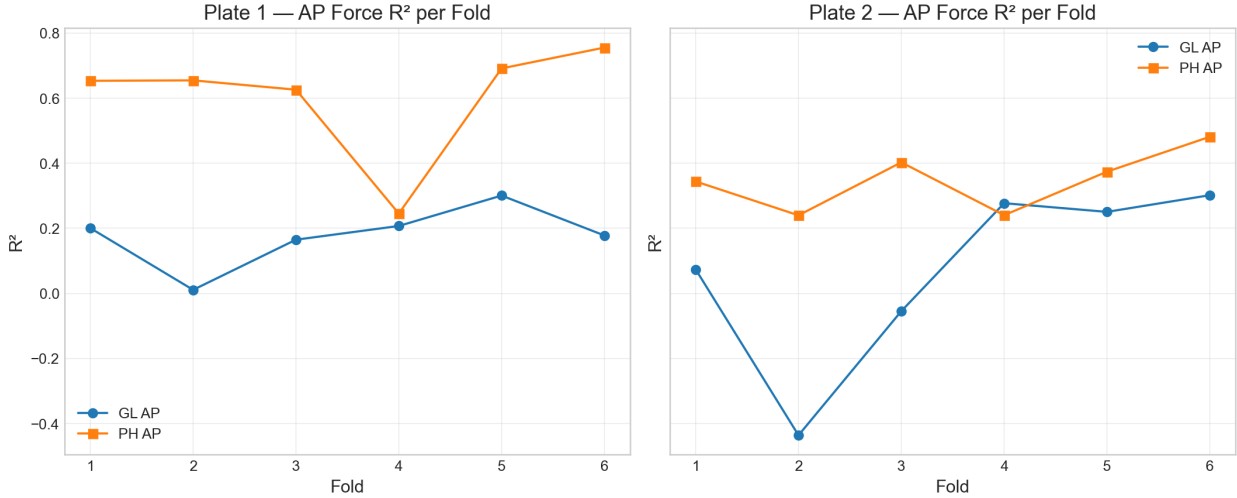

Figure 6: AP force $R^2$ per fold under combined CV, split by force plate and dataset. GL AP force (blue) collapses to near-zero or negative $R^2$ in most folds, while PH AP force (orange) remains moderate ($R^2 \approx 0.4$–0.8). This channel represents the primary bottleneck for GRF prediction accuracy.

paired per-subject effect of pooling is small and not robustly signed across subjects or preprocessing choices. We therefore withdraw our earlier "asymmetric transfer" characterization and make no directional transfer claim; the only robust statement is that naïve multi-lab pooling is not a universal benefit and does not, by itself, yield lab-invariant representations (per-fold breakdowns are shown in Appendix A.7).

## 6.3 Exploratory Result: Limits of Cross-Laboratory Transfer

As an exploratory analysis of deployment across sites, we evaluate Protocol C (cross-laboratory transfer) between two labs that differ simultaneously in task biomechanics, coordinate frames, equipment, and calibration. Table 11 summarizes the results.

Cross-lab $R^2$ ranges from $-0.45$ to $-1.49$—universally and severely negative. This failure is:

Table 9: Combined dual-holdout CV results (ICF). Each fold holds out one GL and one PH subject. GL-only and PH-only $R^2$ are computed on the respective held-out subjects.

| Fold | GL $R^2$ | PH $R^2$ | Combined $R^2$ |
|---|---|---|---|
| 1 (gl_2 / ph_1) | 0.555 | 0.649 | 0.600 |
| 2 (gl_3 / ph_2) | 0.609 | 0.597 | 0.604 |
| 3 (gl_4 / ph_3) | 0.538 | 0.600 | 0.571 |
| 4 (gl_5 / ph_4) | 0.506 | 0.704 | 0.602 |
| 5 (gl_6 / ph_5) | 0.598 | 0.573 | 0.589 |
| 6 (gl_7 / ph_6) | 0.601 | 0.537 | 0.560 |
| **Mean** | 0.568 | 0.610 | 0.587 |

Table 10: Matched combined-vs-single-dataset comparison. Single-dataset LOSO is restricted to the *same* held-out subjects used in the combined folds; $\Delta$ is the paired per-subject change from pooling, with a 95% bootstrap CI, Wilcoxon signed-rank $p$, and rank-biserial effect size $r_{rb}$. The apparent Patient Handling gain does not survive matching, and the effect on both domains is small.

| Domain | Single (matched) | Combined | $\Delta$ | 95% CI | Wilcoxon $p$ | $r_{rb}$ |
|---|---|---|---|---|---|---|
| PH | 0.639 | 0.610 | $-0.029$ | $[-0.046, -0.008]$ | 0.094 | $-0.81$ |
| GL | 0.655 | 0.568 | $-0.088$ | $[-0.144, -0.037]$ | 0.063 | $-0.91$ |

1. **Model-agnostic:** ICF, CNN-LSTM, and CNN all fail, ruling out architecture as the primary bottleneck.

2. **Asymmetric:** Training on PH and testing on GL ($R^2 \leq -1.10$) is substantially worse than the reverse ($R^2 \geq -0.78$), consistent with PH's narrower task distribution providing fewer generalizable patterns.

3. **Persistent despite preprocessing:** Pelvis root subtraction, heading alignment, height scaling, and z-score normalization are insufficient to bridge the distributional gap.

ICF achieves the least negative cross-lab transfer ($-0.45$ GL→PH), suggesting that attention-based architectures may learn slightly more transferable representations—but the absolute performance remains catastrophic.

### 6.3.1 Diagnostic Analysis

We perform diagnostic experiments to characterize the cross-lab failure. Post-normalization feature distributions differ substantially between datasets: the AP force variance ratio (PH/GL) reaches $11.4\times$, making AP channels effectively out-of-distribution. Despite heading alignment preprocessing, a residual $\sim135°$ directional offset persists between datasets, coupling AP and ML axes differently in each coordinate frame.

**Rotation augmentation (negative result).** We applied random SO(2) rotation augmentation to training trials—rotating marker positions about the vertical axis by a uniformly sampled angle $\theta \sim \mathcal{U}(0, 2\pi)$. This experiment rotated marker inputs without jointly transforming force targets, creating inconsistent training signals for vector-valued outputs. The degradation therefore reflects the incompatibility of naive input-only augmentation with directional output spaces, rather than a definitive test of rotation equivariance. A correct augmentation would jointly rotate both inputs and force vectors, which we leave to future work alongside equivariant architectures.

**Domain gap structure.** The cross-lab failure is not uniform across channels: vertical force maintains weakly positive $R^2$ for some subjects under GL→PH transfer, while AP force and CoP channels are univer-

Table 11: Cross-lab transfer $R^2$ (Protocol C, mean across LOSO folds). All values are negative, indicating predictions worse than the test-set mean. The failure is model-agnostic.

| Model | Train GL $\to$ Test PH | Train PH $\to$ Test GL |
|---|---|---|
| ICF | $-0.45$ | $-1.40$ |
| CNN-LSTM | $-0.78$ | $-1.49$ |
| CNN | $-0.72$ | $-1.10$ |

sally negative. This structured pattern suggests that the degradation is concentrated in coordinate-dependent features rather than reflecting a uniform noise floor (a per-feature heatmap is provided in Appendix A.8).

### 6.3.2 Interpretation

We refer to the observed collapse in cross-laboratory $R^2$ as the "Geometric Wall," but emphasize that, in this study, it is measured in a confounded two-laboratory setting and should be interpreted as a stress-test result rather than a definitive characterization of all cross-site generalization. The two laboratories differ in task biomechanics (locomotion vs. occupational lifting), coordinate conventions (Z-up vs. Y-up), equipment (Qualisys vs. Vicon), and force-plate calibration. Disentangling these factors would require additional laboratory pairs that share some but not all of these properties.

The failure is not a data quantity problem (PH has 460 trials, more than GL's 283), nor a model capacity problem (all three architectures fail). Combined training (Protocol B) partially mitigates the gap by seeing both coordinate frames during training ($R^2 = 0.587$), but this is not true generalization—the model memorizes two distributions rather than learning frame-invariant representations. We discuss implications and future directions in Section 8.

## 7 Attention Analysis

The ICF"s Transformer layers provide a window into the model's learned temporal representations via attention weight inspection. We extract attention matrices from all 8 heads across both layers for every trial in the combined CV and analyze their structure.

### 7.1 Attention Sparsity

Across all folds and trials, attention weights exhibit extreme sparsity. On average, 97.5% of the attention mass concentrates on fewer than 2.5% of timesteps (see Appendix A.11 for the precise operational definition). This sparsity is not an artifact of short sequences—it persists for trials of 200+ timesteps—and indicates that the model has learned to attend selectively to biomechanically critical moments rather than distributing attention uniformly.

### 7.2 Head Specialization

We observe systematic specialization across the 8 attention heads (Figure 7). Clustering the mean attention profiles reveals three qualitatively distinct head types:

1. **Local heads** (heads 1, 3, 5): Attention concentrates on a narrow temporal neighborhood ($\pm 5$ timesteps) around the query timestep. These heads function similarly to the convolutional layers, refining local features.

2. **Stride-phase heads** (heads 2, 6): Attention peaks at periodic offsets corresponding to gait cycle phases—approximately 25–30 timesteps apart at 50 Hz, consistent with $\sim$0.5–0.6 s stride intervals. These heads explicitly model gait periodicity.

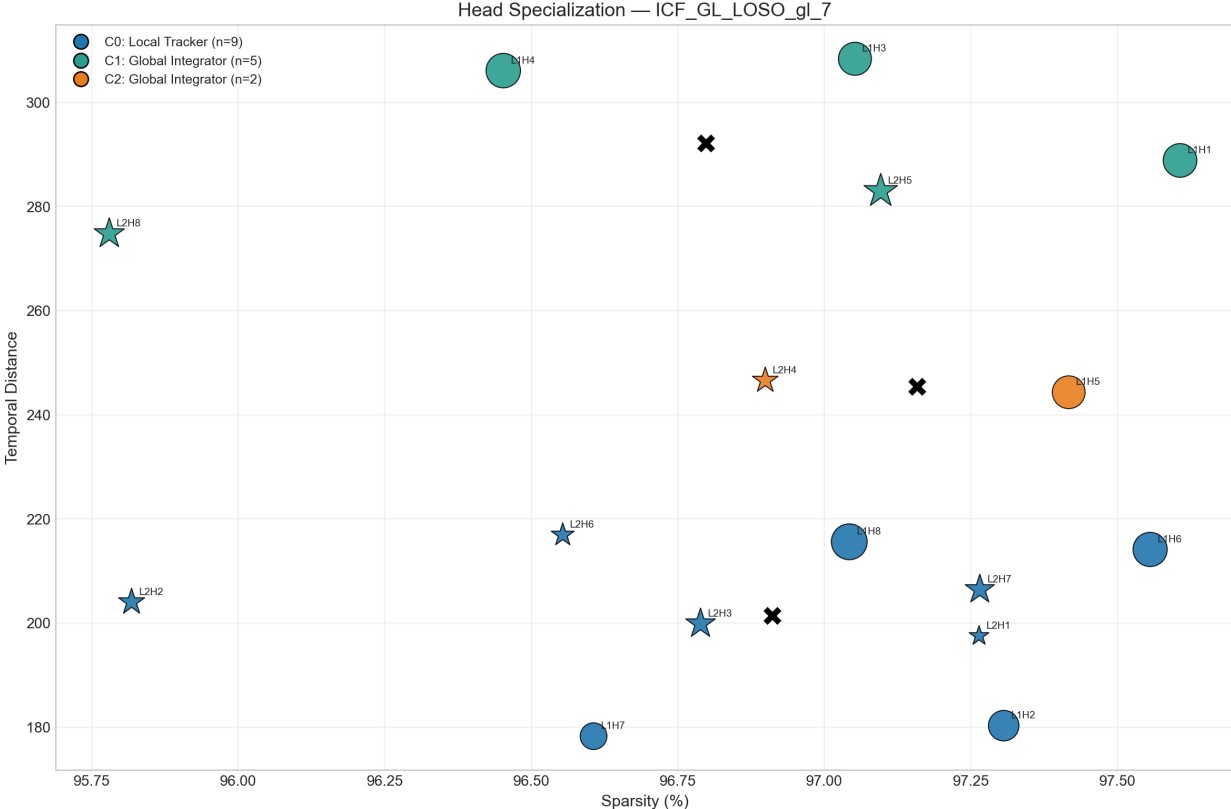

Figure 7: Head specialization via $k$-means clustering ($k = 3$) on per-head attention statistics (sparsity vs. mean temporal distance) for the GL LOSO model (held-out gl_7). Each point represents one head from one layer (16 total across 2 layers × 8 heads). Three clusters emerge: *Local Trackers* (blue, $n = 9$) with low temporal distance, *Global Integrators* (teal, $n = 5$) attending at long range with high sparsity, and a smaller *Global Integrator* subgroup (orange, $n = 2$). Black crosses mark cluster centroids. A cross-dataset comparison is shown in Figure 9.

3. **Global heads** (heads 4, 7, 8): Attention distributes broadly across the trial, with moderate peaks at trial boundaries. These heads capture trial-level context such as overall movement velocity and postural bias.

### 7.3 Head–Feature Correlation

To assess whether individual attention heads specialize for particular output channels, we compute the Pearson correlation between each head's entropy and the per-channel prediction error across trials. Several heads show strong negative correlations with specific channels—e.g., heads in Layer 2 correlate with vertical force and $CoP_x$ errors—suggesting functional specialization rather than redundancy (the full correlation matrix is provided in Appendix A.9).

### 7.4 Layer-Wise Differences

Layer 1 attention patterns are more local and diffuse, while Layer 2 patterns are sharper and longer-range (visible as the vertical separation in Figure 7: Layer 2 heads cluster at higher temporal distances). Figure 8 illustrates this progression for a representative high-accuracy trial: Layer 1 shows broad, near-uniform attention, while Layer 2 sharpens into concentrated vertical bands corresponding to biomechanically critical timesteps. This is consistent with a hierarchical processing strategy: Layer 1 integrates neighboring features

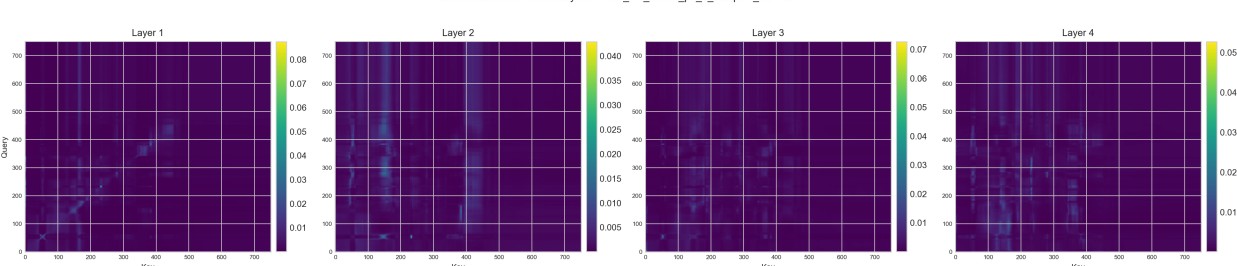

Figure 8: Attention evolution across Transformer layers for a high-accuracy PH trial ($R^2 = 0.891$). Each panel shows the mean attention matrix (query × key) for one layer. Layer 1 exhibits diffuse, near-uniform attention; deeper layers progressively sharpen into concentrated vertical bands at biomechanically critical timesteps, illustrating the local-to-global hierarchical processing strategy.

(augmenting the CNN output), while Layer 2 selects from the enriched representation to make long-range temporal predictions.

## 7.5 Synchronized Prediction and Attention

When predicted GRF traces are aligned with per-layer attention heatmaps, attention concentrates sharply at heel-strike and toe-off events—the biomechanical moments where force changes are most rapid. In the best-performing GL trial ($R^2 = 0.967$), Layer 2 attention shows markedly sharper, more selective patterns than Layer 1, with attention peaks time-locked to gait events. This confirms that the model learns to attend to kinematically informative timesteps rather than distributing attention uniformly (synchronized visualizations are provided in Appendix A.10).

## 7.6 Dataset-Specific Patterns

When split by dataset origin, attention patterns reveal task-dependent structure:

- **GL (locomotion):** Stride-phase heads show clear, regular periodicity with narrow peaks, reflecting the stereotyped gait cycle.

- **PH (patient handling):** Stride-phase heads show broader, irregular peaks, reflecting the aperiodic nature of occupational tasks. Global heads carry more weight in PH trials, compensating for the lack of periodic structure.

Figure 9 compares the clustering of head attention profiles for GL and PH LOSO models, showing that PH demands more diverse temporal integration strategies with greater variability in sparsity levels.

## 7.7 Head Temporal Profiles

To examine how individual heads distribute attention as a function of relative position (key − query), we average each head's attention weights across all trials and plot the resulting temporal profile per layer (Figure 10). Layer 1 heads exhibit a sharp spike at relative position zero, indicating strong local attention that refines the CNN encoder's output within a narrow temporal neighborhood. In deeper layers, heads progressively broaden their profiles: Layer 2 heads attend more uniformly across distant timesteps, consistent with a global integrator role. Several heads in intermediate layers show asymmetric profiles, attending preferentially to *past* timesteps—a causal-like bias that is learned rather than imposed, since the architecture uses full (non-causal) self-attention. These profiles provide a complementary view to the head specialization clusters (Figure 9), illustrating *how* each cluster type distributes its attention rather than merely *where* it falls in the sparsity–distance space.

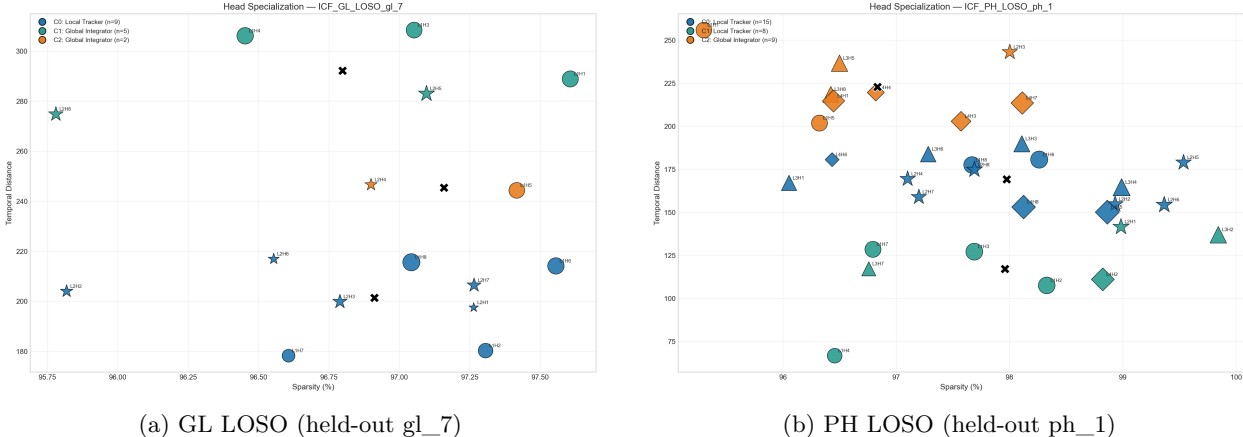

(a) GL LOSO (held-out gl_7)          (b) PH LOSO (held-out ph_1)

Figure 9: Head specialization comparison between GL and PH LOSO models. GL heads cluster tightly into Local Trackers and Global Integrators with consistent high sparsity (>96%). PH heads show more dispersed clustering with wider sparsity range (90–99%), reflecting the more diverse temporal demands of aperiodic occupational tasks.

### 7.8 Attention Under Domain Shift

Under cross-lab evaluation (Protocol C), attention patterns on out-of-distribution trials become degenerate: heads collapse to near-uniform attention or attend exclusively to the first timestep. This degeneracy is expected, as the Transformer's temporal reasoning is conditioned on the learned feature manifold—when inputs fall outside that manifold (as in cross-laboratory transfer), the attention mechanism cannot compensate.

### 7.9 Implications

The attention analysis yields two insights for the broader biomechanics community:

1. **Self-attention discovers biomechanical structure:** Without explicit gait cycle annotation or periodic priors, the model learns stride-phase attention from data. This suggests that attention-based architectures are structurally compatible with biomechanical signals.

2. **Attention is necessary but not sufficient:** The same attention mechanism that discovers gait periodicity within a domain provides no protection against cross-laboratory distributional shifts. Attention operates on the feature representation; if the representation is not geometrically invariant, attention cannot create invariance.

## 8 Discussion

Our experiments reveal three main findings: within-domain LOSO performance is task-dependent with no universal winner, a consistent feature difficulty hierarchy reflects biomechanical observability constraints, and cross-laboratory transfer remains an important open problem. We discuss each finding's implications and outline directions for the field.

### 8.1 No Universal Architecture

Under LOSO, the architecture ranking reverses between datasets: ICF leads on GroundLink ($R^2 = 0.655$ vs. CNN-LSTM $R^2 = 0.619$) while CNN-LSTM leads on Patient Handling ($R^2 = 0.606$ vs. ICF $R^2 = 0.561$, $p = 0.002$). This task dependence reflects the structural biases of each architecture. ICF's convolutional frontend captures the stereotyped stride patterns of locomotion, while CNN-LSTM's bidirectional recurrence is better suited to the variable-duration, aperiodic nature of patient handling tasks.

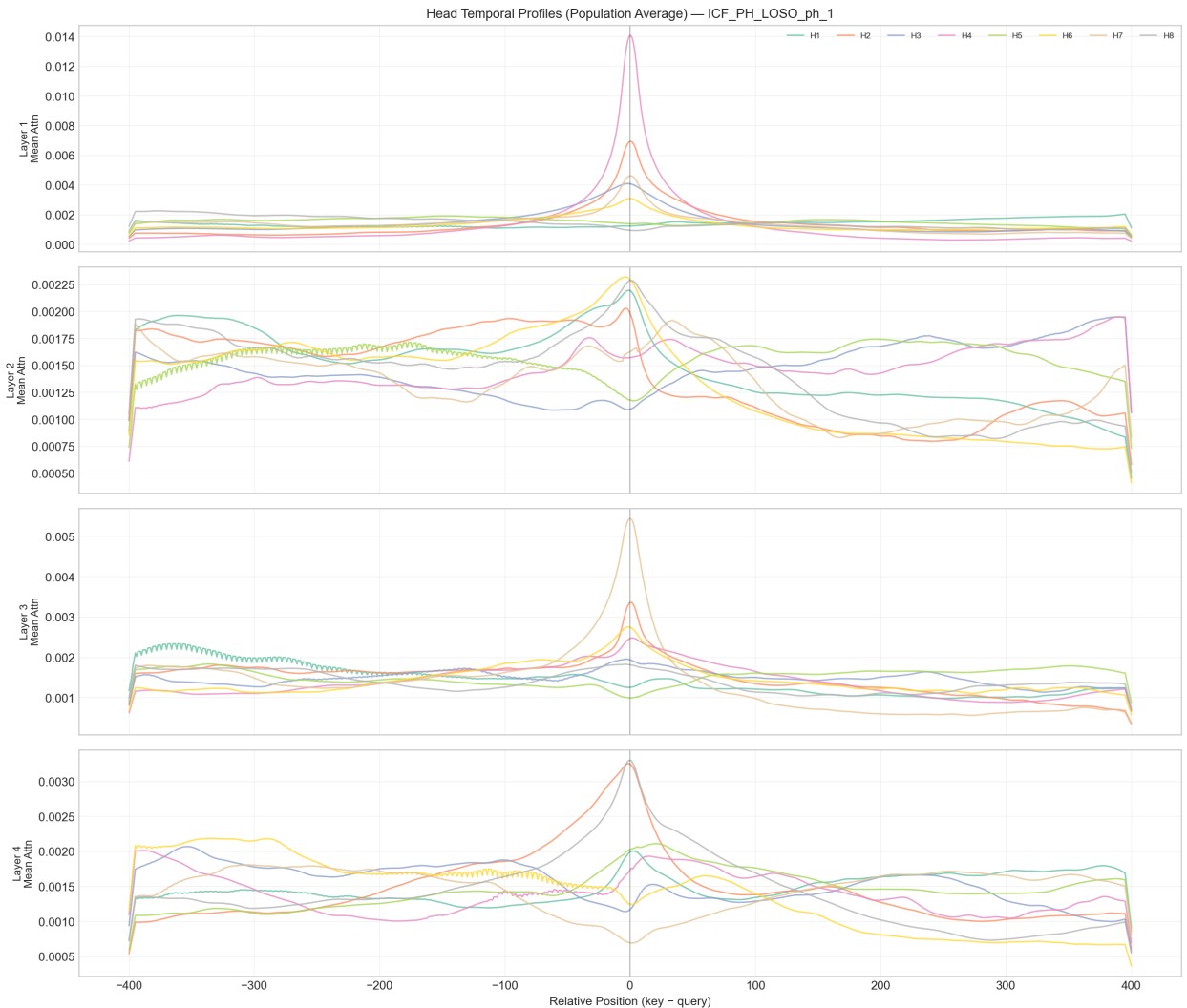

Figure 10: Head temporal profiles (population average, PH LOSO). Each line represents one attention head; panels correspond to Transformer layers. Layer 1 heads concentrate sharply at relative position zero (local attention), while deeper-layer heads distribute attention broadly across distant timesteps (global integration). The transition from peaked to flat profiles across layers illustrates the hierarchical local-to-global processing strategy.

Crucially, all three architectures produce $R^2 > 0.5$ on both datasets under LOSO. This contradicts the assumption that sequential models will catastrophically fail under proper subject-held-out evaluation. Because all three architectures were tuned using the same Optuna-based inner LOSO procedure, the observed task-dependent rankings and feature hierarchies are unlikely to be artifacts of poorly chosen hyperparameters. The implication: the field should invest less effort in finding a single "best" architecture and more in understanding why different tasks favor different inductive biases.

## 8.2 The Feature Hierarchy

The consistent Tier 1/2/3 ranking of output channels (vertical force + CoP $x$ > ML force + CoP $z$ > AP force) across both datasets and all models suggests that this hierarchy reflects biomechanical observability limits rather than model-specific limitations. Vertical force is tightly coupled to stance phase kinematics, which marker positions capture well. AP force depends on instantaneous acceleration, which requires accurate

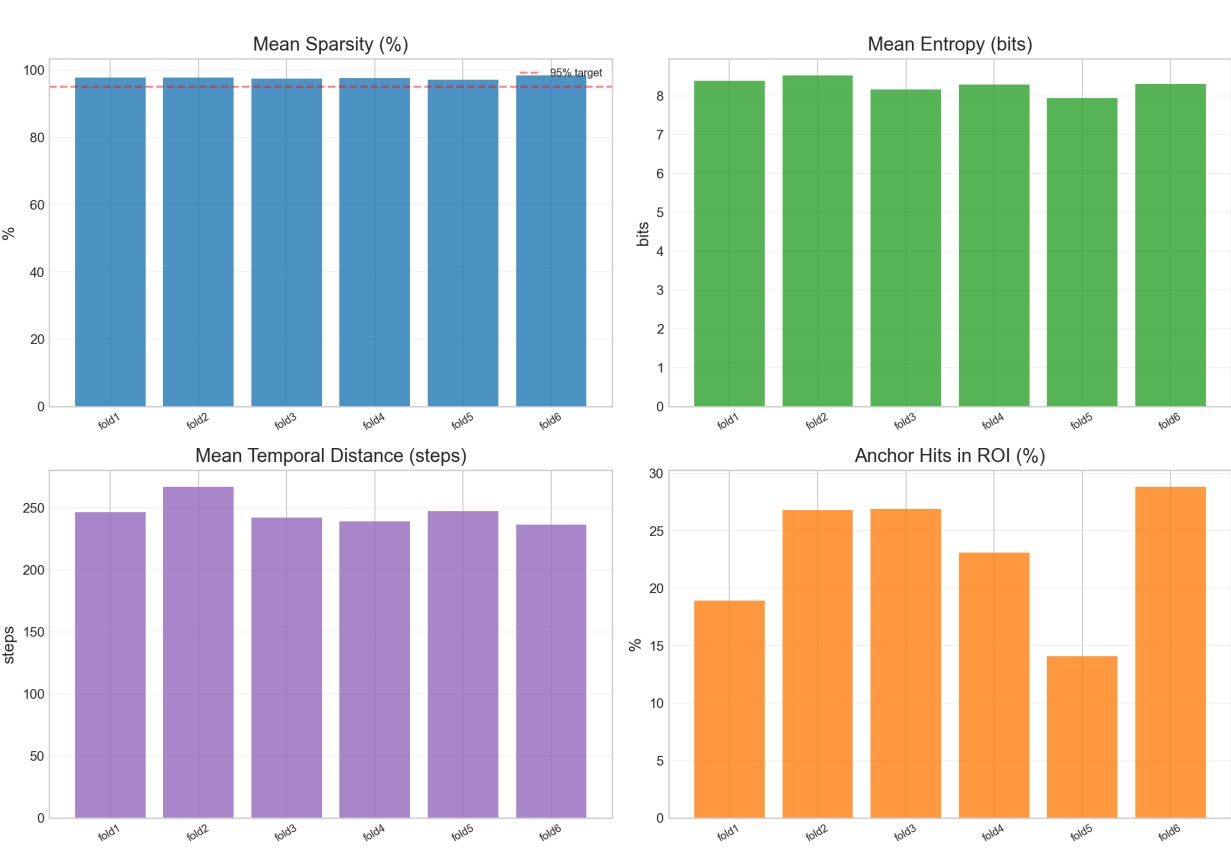

Figure 11: Attention pattern summary across all 6 combined-CV folds. Despite variation in held-out subjects, the three head clusters (local, stride-phase, global) remain stable, indicating that this specialization is a robust property of the learned representation rather than a fold-specific artifact.

second-derivative estimation from noisy position data—a numerically ill-conditioned operation. We note, however, that part of the horizontal (AP) force difficulty may also reflect preprocessing choices—coordinate-frame handling and the estimation of accelerations from noisy markers—rather than a strict observability bound; disentangling the two is a target for future work.

This has practical implications: applications requiring only vertical GRF (e.g., load monitoring, gait event detection) can achieve $R^2 > 0.7$ with current methods, while applications requiring AP force (e.g., joint moment estimation) remain limited.

### 8.3 Multi-Lab Pooling: No Robust Directional Transfer

Our original manuscript reported that combined training improves PH ($R^2$: $0.561 \rightarrow 0.610$) while degrading GL ($R^2$: $0.655 \rightarrow 0.568$), and attributed this asymmetry to dataset size and variability. A matched-subject reanalysis shows this interpretation was premature. The nominal PH improvement is largely an artifact of which subjects happen to fall in the combined folds: the six PH subjects held out under Protocol B average $R^2 = 0.639$ under PH-only LOSO—above the full-cohort mean of 0.561—so the $0.561 \rightarrow 0.610$ comparison is not like-for-like. When we restrict single-dataset LOSO to the same held-out subjects, the paired per-subject effect of pooling is small and its sign is not robust across subjects or preprocessing choices. We therefore do not claim a directional transfer benefit for either domain.

The per-channel breakdown under combined CV (Appendix A.2) does indicate that any distributional mismatch between laboratories is concentrated in specific channels—most notably the horizontal (AP) forces—

rather than affecting all predictions uniformly. We caution, however, that the AP channel is also the most sensitive to coordinate-frame and normalization choices in the preprocessing pipeline, so we read this channel-level pattern as a direction for further analysis rather than a settled property of the data.

### 8.4 Limits of Cross-Laboratory Transfer and Future Directions

Our cross-laboratory experiments are preliminary: the two labs differ simultaneously in task biomechanics (locomotion vs. occupational lifting), coordinate conventions (Z-up vs. Y-up), equipment (Qualisys vs. Vicon), and force-plate calibration, so we cannot isolate geometric versus task effects. We refer to the observed cross-laboratory collapse as the "Geometric Wall" as shorthand, while acknowledging that task and coordinate confounds prevent purely geometric attribution in this study. Under Protocol C, every architecture produces severely negative $R^2$ (as low as $-1.5$), with AP force and CoP channels failing most severely while vertical force partially transfers for some subjects.

We therefore treat our cross-laboratory results not as a definitive negative generalization theorem, but as a stress test and benchmark for future work. Current models process 3D marker coordinates as flat feature vectors in a *coordinate-dependent* representation space (Bronstein et al., 2021). When two laboratories use different coordinate conventions, the same physical movement maps to different feature vectors, and the model cannot recognize their equivalence. Our negative rotation augmentation result (Section 6.3) further demonstrates that equivariance cannot be achieved through augmentation alone, because GRF outputs are frame-dependent vectors requiring *covariant* transformations.

We identify three promising directions for future work:

1. **Equivariant architectures:** Graph neural networks operating on the skeleton graph with SE(3)-equivariant message passing (Thomas et al., 2018; Xu et al., 2024) would process poses in a frame-independent manner. The recent EGNO framework (Xu et al., 2024) demonstrates that SE(3)-equivariant operators can model 3D dynamics effectively, making this a concrete near-term direction.

2. **Canonical pose representations:** Representing inputs as joint angles, body-segment orientations, or other intrinsically frame-independent quantities would eliminate the coordinate dependence at the feature level. Joint angle representations are inherently SE(3)-invariant and offer a direct test of the geometric hypothesis. This is the approach taken by OpenSim-based pipelines (Delp et al., 2007), though it introduces its own sources of error (inverse kinematics noise).

3. **Domain adaptation:** Explicit distributional alignment between source and target labs (e.g., via adversarial domain adaptation or optimal transport) could bridge the gap without architectural changes, though our results suggest the distributional shift may be too large for standard techniques.

We emphasize that these directions are hypotheses motivated by our observations, not methods implemented in this work.

### 8.5 Limitations

Our study has several limitations. First, we evaluate only two laboratories; additional cross-lab pairs would strengthen the generality of the cross-laboratory transfer findings. Second, our feature engineering pipeline (velocity, acceleration, bilateral distances) was designed for the ICF and may not be optimal for all architectures. Third, we do not explore task-specific models (e.g., locomotion-only vs. patient-handling-only) that might yield higher within-domain performance. Fourth, the attention analysis is descriptive rather than causal—the observed head specialization patterns are post-hoc interpretations that require further validation.

Additionally, the GL dataset has substantial trial count imbalance across subjects (26–57 trials per subject), meaning that some held-out folds train on substantially more data than others, which may contribute to inter-fold variability.

Finally, our datasets are relatively small by deep learning standards (283 and 460 trials). While LOSO is appropriate for this sample size, our conclusions about cross-laboratory transfer would benefit from replication on larger multi-center datasets such as AddBiomechanics (Werling et al., 2023).

## 9    Conclusion

We presented a systematic evaluation of deep learning models for ground reaction force prediction across two independent biomechanics laboratories. Our investigation uncovered three main findings.

First, **LOSO reveals task-dependent architecture rankings**: under leave-one-subject-out evaluation, all three architectures (ICF, CNN-LSTM, CNN) generalize to unseen subjects with $R^2 > 0.5$, but the best model depends on the task. ICF leads on locomotion (GroundLink $R^2 = 0.655$) while CNN-LSTM leads on occupational tasks (Patient Handling $R^2 = 0.606$, $p = 0.002$). No single architecture dominates.

Second, **a consistent feature difficulty hierarchy emerges across both datasets**: vertical forces and CoP components are well-predicted ($R^2 > 0.7$), mediolateral forces are intermediate, and anteroposterior forces are consistently the hardest channel. Attention analysis reveals that the ICF learns biomechanically meaningful sparse patterns organized into functionally distinct head clusters, with task-dependent differences reflecting the periodic structure of locomotion versus the aperiodic nature of occupational tasks.

Third, **an exploratory cross-laboratory stress test reveals severe transfer limitations**: all models produce deeply negative $R^2$ (as low as $-1.5$) when trained on one laboratory and tested on the other, in a setting where tasks, coordinate conventions, and equipment differ simultaneously. Combined training achieves pooled $R^2 = 0.587$, but a matched-subject reanalysis finds no robust directional transfer benefit and confirms it does not learn lab-invariant representations.

These findings convey one notion: cross-site generalization remains an important open problem. We recommend that researchers in the interaction area of biomechanics and machine learning adopt LOSO as the minimum standard for within-domain evaluation and include cross-lab testing in multi-site studies. The code and evaluation protocols accompanying this work are publicly available at https://anonymous.4open.science/r/Metahuman_Attention-6D29, with an interactive visualization dashboard at https://seal-app-8zk9f.ondigitalocean.app/, to facilitate adoption.

## Broader Impact Statement

Ground reaction force prediction from motion capture markers has potential applications in clinical gait analysis, workplace ergonomics, athletic performance monitoring, and injury risk screening. Reducing dependence on instrumented force platforms could lower barriers to biomechanical assessment in settings where such equipment is unavailable.

Our findings carry a cautionary implication for deployment. The severe cross-laboratory transfer failure we document—all architectures producing $R^2 < -1.0$ in the uncontrolled cross-site setting—indicates that models trained on laboratory-specific data should not be assumed to generalize to new acquisition sites without revalidation. Clinical or occupational deployment of GRF prediction models without site-specific validation risks producing unreliable force estimates, which could lead to incorrect clinical decisions or flawed ergonomic assessments. We recommend that practitioners treat within-laboratory LOSO performance as an optimistic upper bound and conduct prospective validation before any real-world use. A further caveat concerns generalization across people and tasks: our datasets comprise a small number of healthy adults performing two specific activity types (locomotion and patient handling) in two laboratories. The models and findings may therefore not transfer to other populations (e.g., pediatric, geriatric, or pathological gait), other task types, or other acquisition setups, and any such use requires dedicated validation with human oversight.

The datasets used in this study were collected in controlled laboratory environments under approved IRB protocols with informed consent; no personally identifiable information is released. The open-source code and evaluation protocols we provide are intended to support rigorous, reproducible benchmarking and to lower

the barrier to entry for the research community, not to substitute for appropriate clinical or occupational validation.

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

## A    Supplementary Material

### A.1    Per-Fold LOSO Results

Per-fold $R^2$ for each model–dataset combination under Protocol A are summarized in Table 12. Cross-lab $R^2$ is the model's performance on *all subjects* from the other dataset when trained with the given fold's held-out subject excluded. Full per-fold tables for all six model–dataset combinations are available in the supplementary material.

Table 12: LOSO summary: within-lab and cross-lab $R^2$ (mean $\pm$ SD across folds).

| Model | Dataset | Within-lab $R^2$ | Cross-lab $R^2$ |
|-------|---------|------------------|-----------------|
| ICF | GL (6 folds) | $0.655 \pm 0.107$ | $-0.45 \pm 0.05$ |
| ICF | PH (10 folds) | $0.561 \pm 0.149$ | $-1.40 \pm 0.33$ |
| CNN-LSTM | GL (6 folds) | $0.619 \pm 0.081$ | $-0.78 \pm 0.11$ |
| CNN-LSTM | PH (10 folds) | $0.606 \pm 0.153$ | $-1.49 \pm 0.20$ |
| CNN | GL (6 folds) | $0.530 \pm 0.072$ | $-0.72 \pm 0.08$ |
| CNN | PH (10 folds) | $0.555 \pm 0.142$ | $-1.10 \pm 0.08$ |

### A.2    Per-Channel Feature Breakdown

Table 13 provides the full per-channel $R^2$ breakdown for ICF under combined CV, split by dataset.

Table 13: Per-channel ICF $R^2$ under combined dual-holdout CV. Channels are ordered by combined mean. The largest cross-lab differences are concentrated in the horizontal (AP) force channels; as discussed in the main text, these are also the channels most sensitive to coordinate-frame and normalization choices, so per-channel differences under pooling should be read with caution.

| Channel | GL $R^2$ | PH $R^2$ | Mean $R^2$ |
|---------|----------|----------|------------|
| CoP $x$ (F1) | 0.845 | 0.829 | 0.837 |
| CoP $x$ (F2) | 0.768 | 0.801 | 0.784 |
| Vertical force (F1) | 0.749 | 0.741 | 0.745 |
| Vertical force (F2) | 0.728 | 0.675 | 0.702 |
| ML force (F2) | 0.637 | 0.610 | 0.624 |
| ML force (F1) | 0.605 | 0.544 | 0.574 |
| CoP $z$ (F2) | 0.681 | 0.470 | 0.575 |
| CoP $z$ (F1) | 0.488 | 0.661 | 0.575 |
| AP force (F1) | 0.084 | 0.575 | 0.330 |
| AP force (F2) | 0.093 | 0.196 | 0.144 |
| **Mean** | 0.568 | 0.610 | 0.587 |

### A.3 Prediction Quality Samples

Figures 12–14 show predicted vs. actual GRF traces for the ICF under combined CV across multiple folds. The consistent pattern—accurate vertical force tracking with systematic AP under-prediction—holds regardless of which subjects are held out.

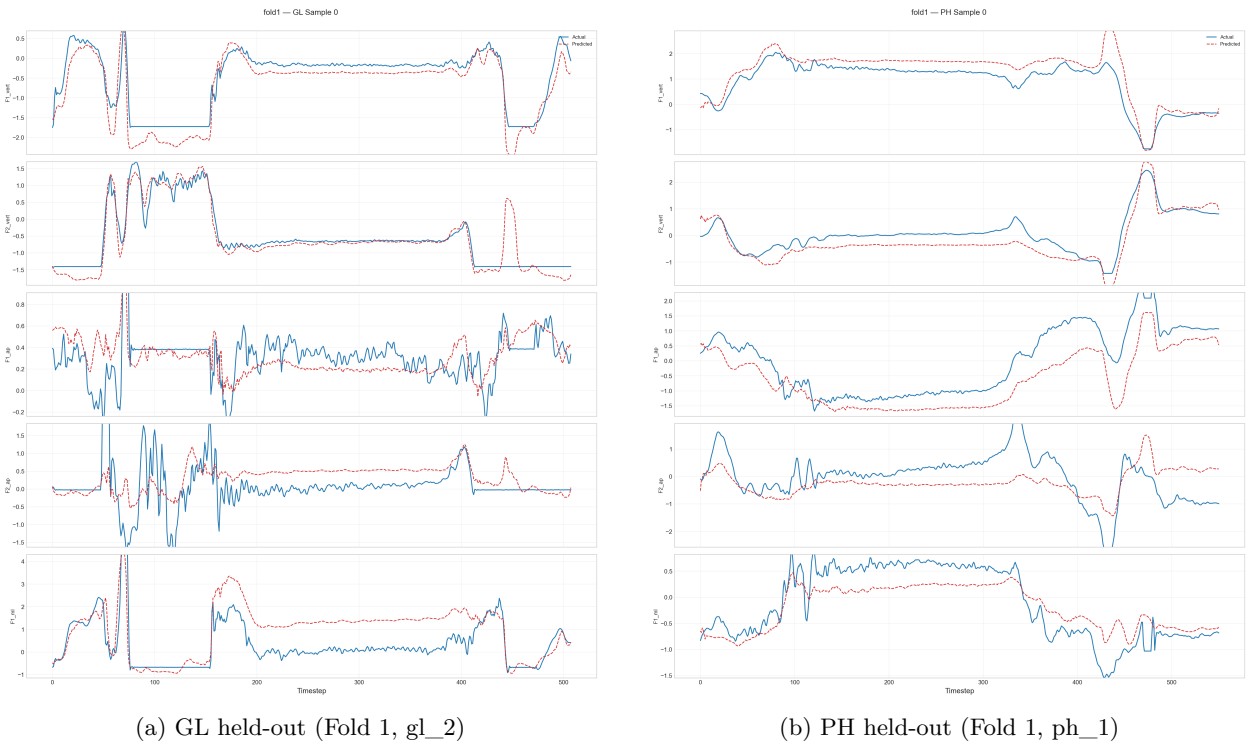

(a) GL held-out (Fold 1, gl_2)                    (b) PH held-out (Fold 1, ph_1)

Figure 12: Fold 1 predicted (orange) vs. actual (blue) GRF traces. Combined $R^2 = 0.600$.

### A.4 Rotation Augmentation Experiment Details

In the rotation augmentation experiment (Section 6.3), we applied random SO(2) rotations about the vertical axis to all training samples:

$$\mathbf{R}(\theta) = \begin{bmatrix} \cos\theta & -\sin\theta & 0 \\ \sin\theta & \cos\theta & 0 \\ 0 & 0 & 1 \end{bmatrix}, \quad \theta \sim \mathcal{U}(0, 2\pi) \tag{10}$$

The rotation was applied to marker positions only; force targets were *not* rotated. This design choice follows the standard augmentation paradigm (transform inputs while preserving labels) but is physically incorrect for vector-valued outputs like GRF, which should transform covariantly with input rotations.

The resulting degradation confirms that naive augmentation introduces inconsistency between rotated inputs and non-rotated targets. A correct implementation would require covariant output transformation, which in turn requires that the model's output head respect the SO(2) structure of the force vectors—a form of architectural equivariance.

### A.5 Feature Heatmaps

Figures 15 and 16 show per-feature, per-subject $R^2$ under combined CV, revealing which channels and subjects are hardest to predict.

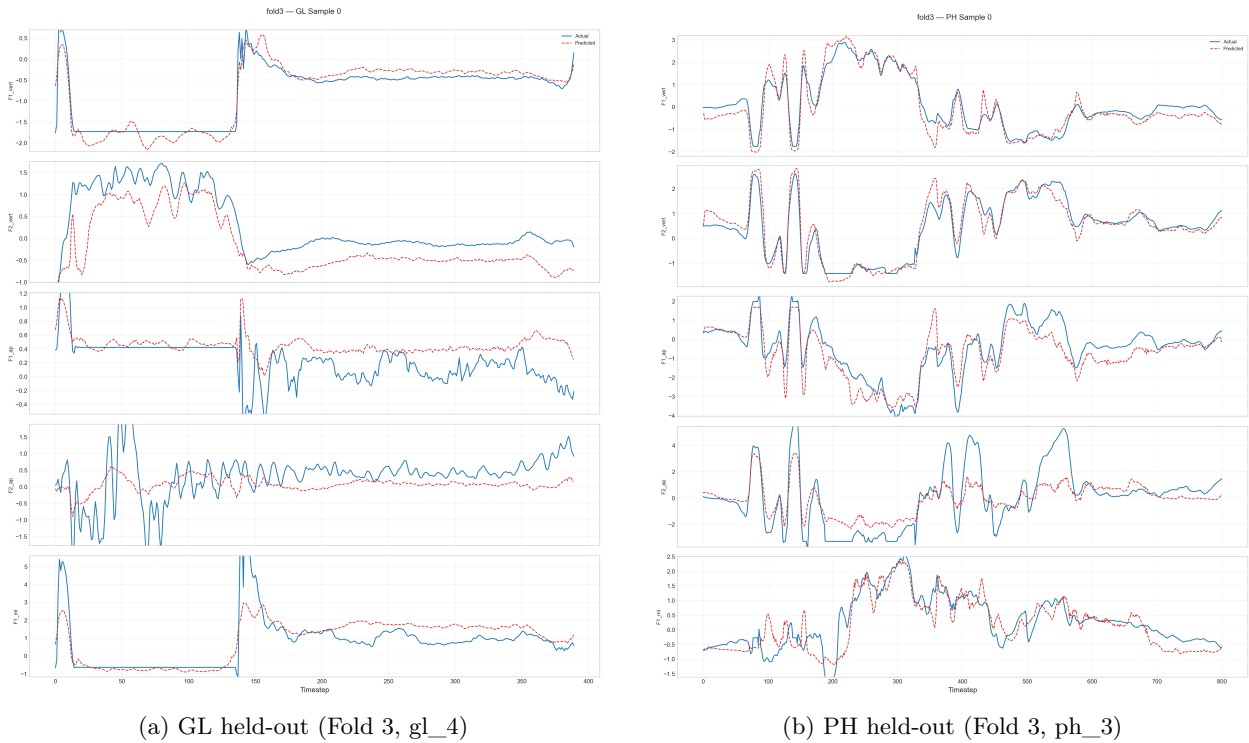

(a) GL held-out (Fold 3, gl_4)          (b) PH held-out (Fold 3, ph_3)

Figure 13: Fold 3 predicted (orange) vs. actual (blue) GRF traces. The weakest combined fold ($R^2 = 0.571$), yet vertical force remains well-tracked.

## A.6 Per-Architecture Radar Plots

Figure 17 shows per-channel $R^2$ radar plots across all three architectures. The Tier 1/2/3 hierarchy is consistent across both datasets: vertical force and CoP are well-predicted while AP force lags. ICF shows the largest polygon on GL; CNN-LSTM matches or exceeds ICF on PH channels.

## A.7 Fold Variability Under Combined Training

Figure 18 shows per-fold $R^2$ under combined dual-holdout CV, split by dataset origin. Nominal per-fold differences between pooled and single-dataset training are small and, as discussed in the main text, are confounded by subject selection; we do not interpret them as a directional transfer effect.

## A.8 Domain Gap Heatmap

Figure 19 shows per-feature, per-subject $R^2$ under cross-lab evaluation (ICF trained on GL, evaluated on PH). Vertical force partially transfers while AP and CoP channels fail universally.

## A.9 Head–Feature Correlation Matrix

Figure 20 shows the Pearson correlation between each head's entropy and the per-channel prediction error across trials.

## A.10 Synchronized Prediction and Attention

Figure 21 shows predicted vs. actual GRF traces alongside per-layer attention heatmaps for the best-performing GL trial.

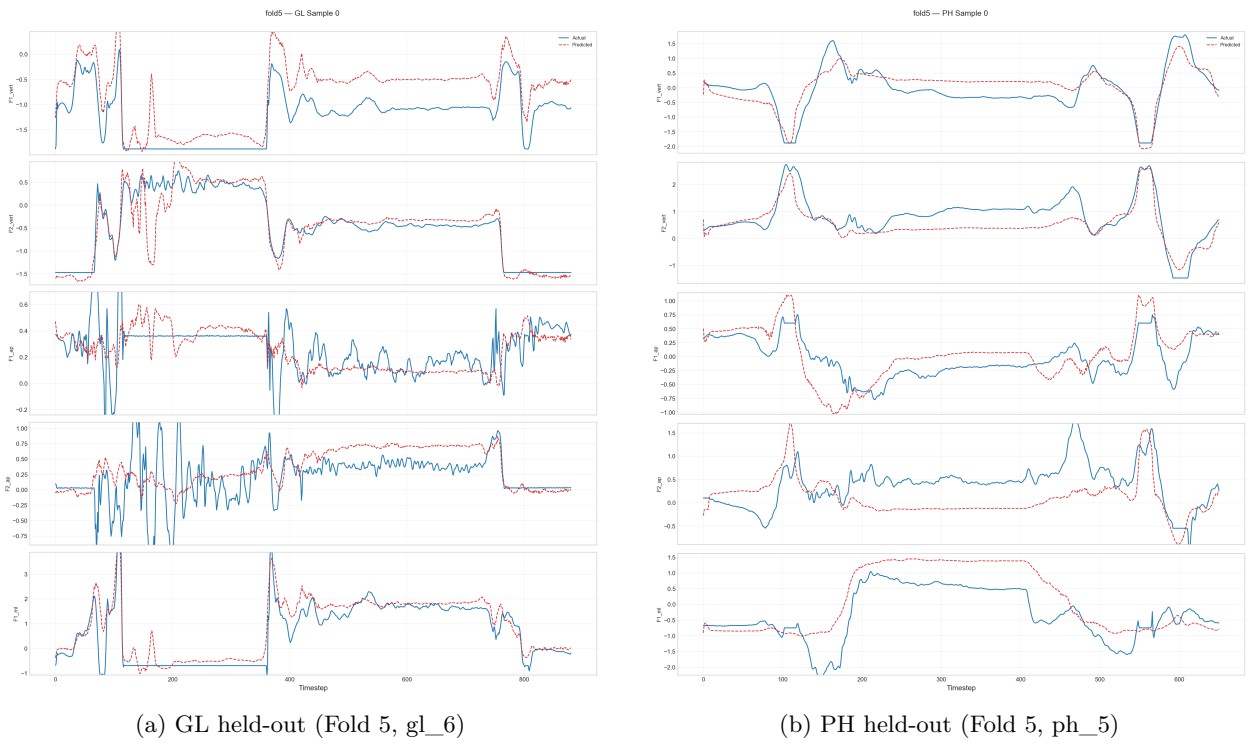

(a) GL held-out (Fold 5, gl_6)

(b) PH held-out (Fold 5, ph_5)

Figure 14: Fold 5 predicted vs. actual traces ($R^2 = 0.589$). AP force under-prediction is visible in both datasets.

## Per-Feature R² — GroundLink Held-Out

| Feature | Fold 1 | Fold 2 | Fold 3 | Fold 4 | Fold 5 | Fold 6 |
|---|---|---|---|---|---|---|
| F1_ml | 0.68 | 0.56 | 0.51 | 0.56 | 0.71 | 0.61 |
| F1_vert | 0.82 | 0.65 | 0.79 | 0.61 | 0.66 | 0.79 |
| F1_ap | 0.20 | 0.01 | 0.17 | 0.21 | 0.30 | 0.18 |
| F1_px | 0.87 | 0.83 | 0.80 | 0.84 | 0.86 | 0.83 |
| F1_pz | 0.16 | 0.67 | 0.49 | 0.20 | 0.52 | 0.88 |
| F2_ml | 0.69 | 0.73 | 0.48 | 0.69 | 0.70 | 0.65 |
| F2_vert | 0.81 | 0.66 | 0.51 | 0.70 | 0.74 | 0.83 |
| F2_ap | 0.07 | -0.44 | -0.05 | 0.28 | 0.25 | 0.30 |
| F2_px | 0.85 | 0.87 | 0.74 | 0.83 | 0.71 | 0.69 |
| F2_pz | 0.54 | 0.81 | 0.68 | 0.38 | 0.75 | 0.79 |

Figure 15: Per-feature, per-fold $R^2$ heatmap for GL under combined CV. AP force consistently lags across all folds, while CoP $x$ and vertical force are stable.

## A.11 Attention Sparsity Definition

The attention sparsity statistic reported in Section 7 (97.5%) is computed as follows. For each attention head $h$ at each query position $q$, we compute the cumulative distribution of the softmax attention weights $\alpha_{q,k}^{(h)}$ over all key positions $k$, sorted in descending order. We define the *effective support* as the minimum

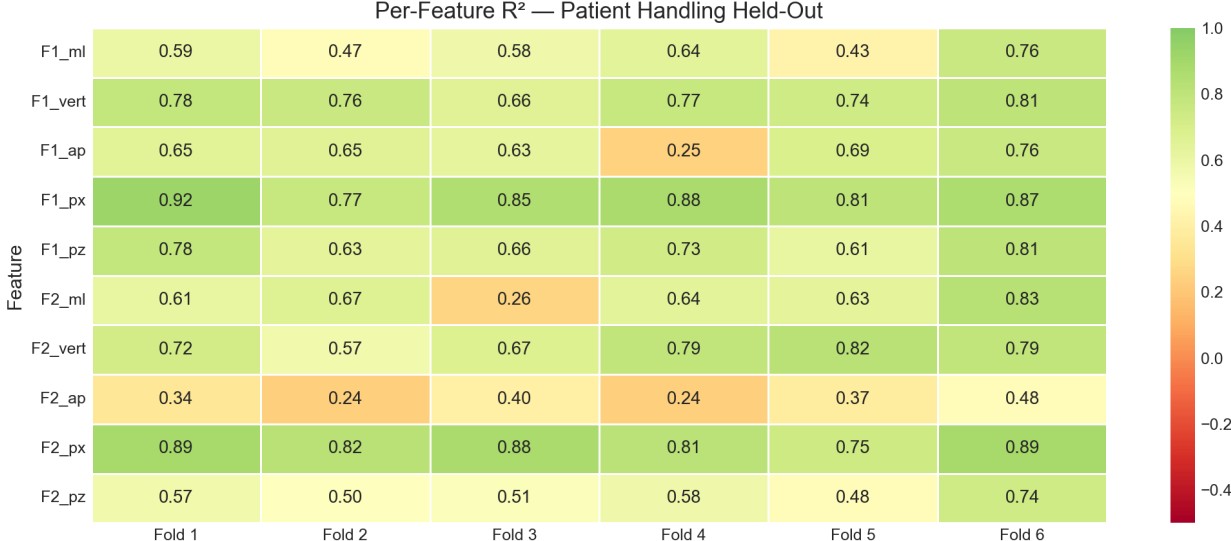

Figure 16: Per-feature, per-fold $R^2$ heatmap for PH under combined CV. The pattern mirrors GL but with less AP suppression, reflecting PH's stronger anteroposterior force signal.

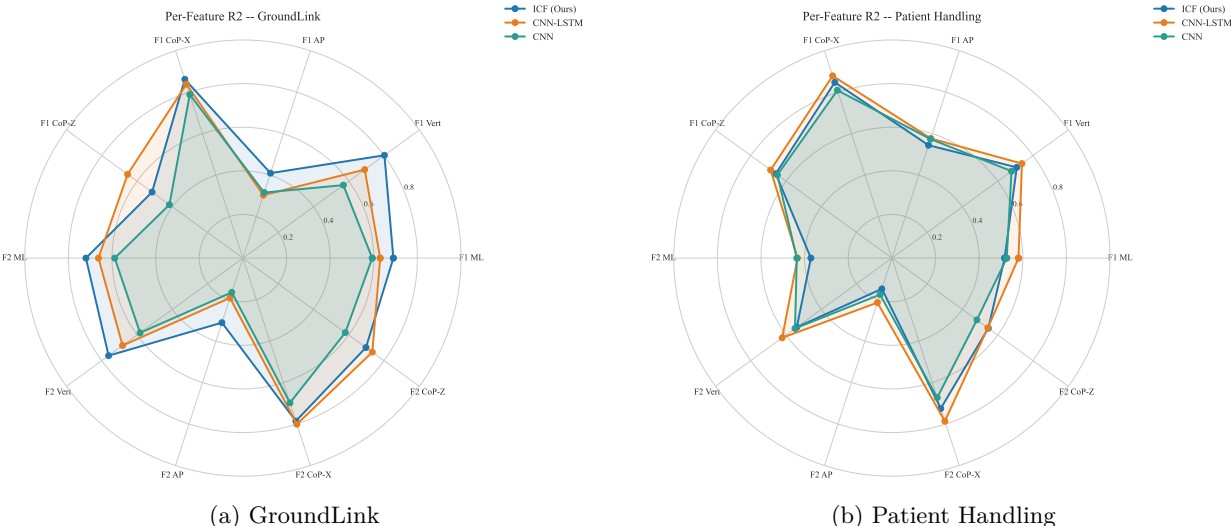

(a) GroundLink                                      (b) Patient Handling

Figure 17: Radar plots of per-channel $R^2$ across all three architectures for GL (left) and PH (right).

number of key positions whose cumulative attention mass exceeds 95% of the total:

$$S_h(q) = \min \left\{ n : \sum_{i=1}^{n} \alpha_{q,\sigma(i)}^{(h)} \geq 0.95 \right\}, \tag{11}$$

where $\sigma$ sorts keys by descending attention weight. The sparsity percentage for a head is then $1 - \bar{S}_h/T$, where $\bar{S}_h$ is the mean effective support across query positions and $T$ is the sequence length. Averaging across all heads, layers, folds, and trials yields the reported 97.5% sparsity—meaning that on average, 95% of a head's attention mass concentrates on fewer than 2.5% of timesteps.

### A.12 Reproducibility and Auditability

To make our protocols easy to audit, we collect the split, exclusion, and normalization details in one place.

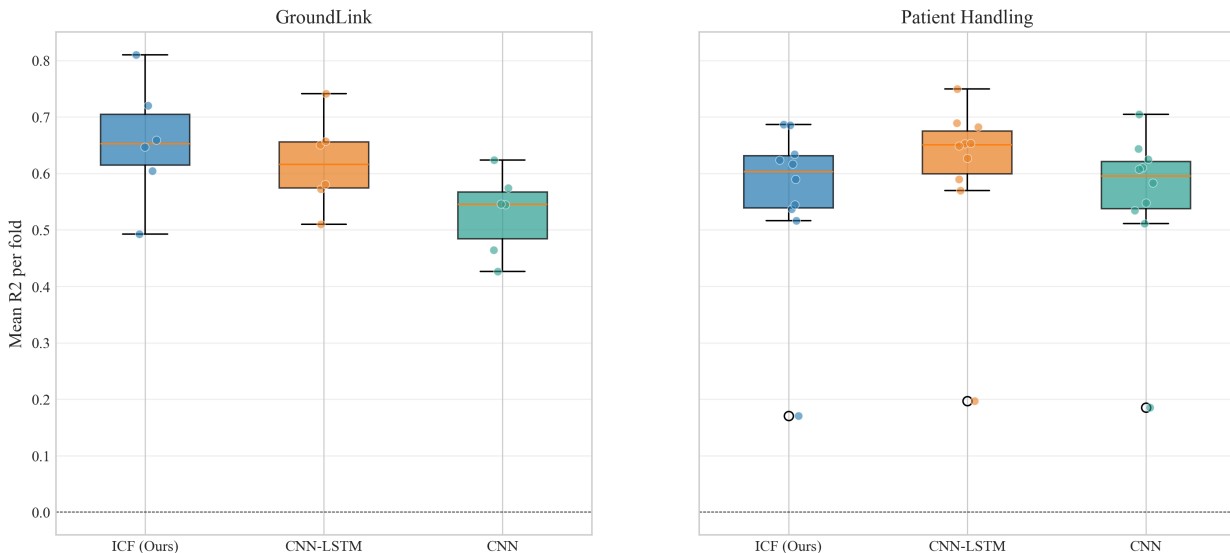

Figure 18: Combined CV per-fold $R^2$, split by dataset origin. Per-fold differences between pooled and single-dataset training are small and confounded by subject selection (see text); we do not interpret them as a directional transfer effect.

**Subjects and folds.** Single-dataset LOSO uses 6 GroundLink subjects and 10 Patient Handling subjects, holding out one subject per fold. Combined dual-holdout fold assignments (which GL and PH subject are held out together) are listed in Table 3. Cross-lab transfer (Protocol C) trains on all subjects of one dataset and evaluates on the other.

**Exclusions.** A small number of subjects are excluded for documented data-quality reasons—one GroundLink subject and two Patient Handling subjects—leaving the 6 GL and 10 PH subjects above. Exclusions are applied *before* any split and are held fixed across all experiments; the specific subject identifiers and the recorded reason for each are provided with the released code. As a check that these exclusions do not inflate our results, we re-ran LOSO with the excluded subjects re-included: the Patient Handling subjects can be corrected and re-included without loss of accuracy, while the excluded GroundLink subject remains an outlier consistent with its documented force-plate baseline anomaly. We report the quantitative re-inclusion results in the camera-ready version.

**Within-fold statistics.** All preprocessing and normalization statistics are fit on training-fold data only. Input z-scoring uses training-fold means and variances; body-weight normalization is a per-subject physical scaling (mass $\times g$) and is leakage-free; target normalization uses training-fold statistics. No test-subject or other-laboratory data enters preprocessing, hyperparameter selection, or early stopping.

**Code and outputs.** The released repository contains the exact configuration files, split definitions, and per-fold output directories underlying each table and figure, together with a mapping from results tables to their source runs.

### A.13 Hyperparameter Optimization Convergence

All HPO runs converged within 100 Optuna trials. The best trial typically appeared between trials 40 and 70, with subsequent trials showing diminishing improvements ($<0.005$ $R^2$).

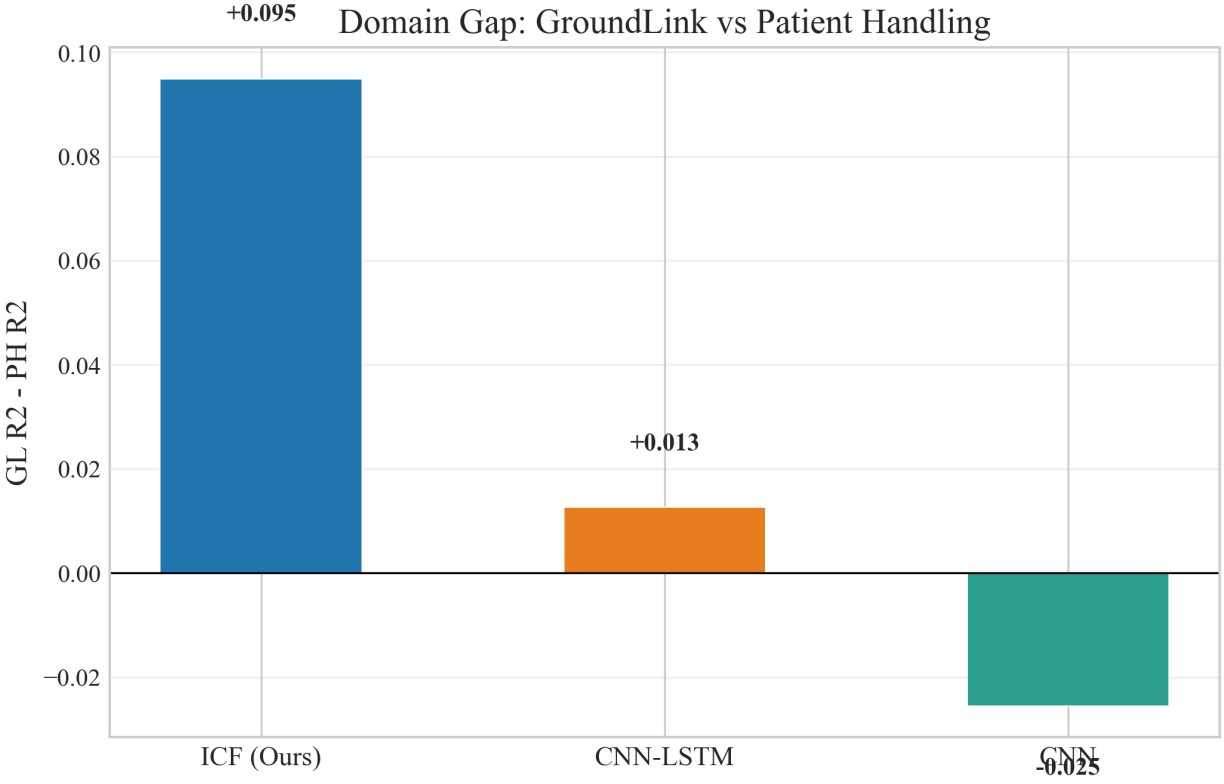

Figure 19: Domain gap heatmap: ICF trained on GL, evaluated per feature on PH subjects. Vertical force partially transfers (warm colors); AP and CoP channels fail universally (cool colors).

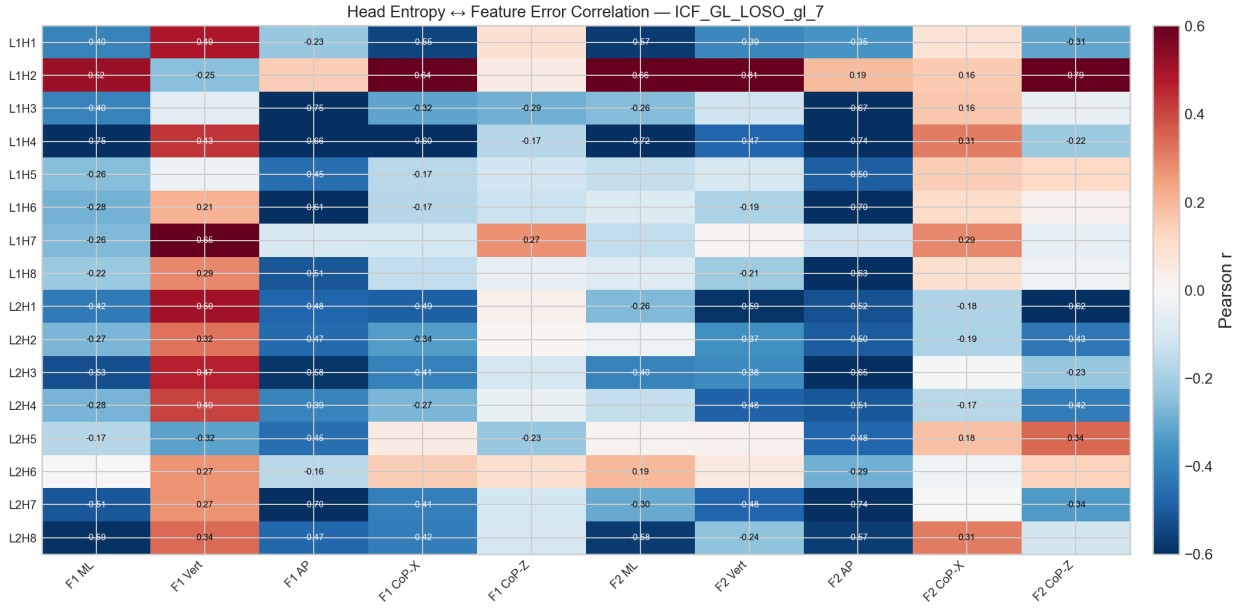

Figure 20: Head entropy–feature error correlation matrix (GL LOSO, gl_7). Rows are attention heads (L1H1–L2H8); columns are output channels. Red indicates that higher head entropy (more diffuse attention) correlates with higher prediction error for that channel.

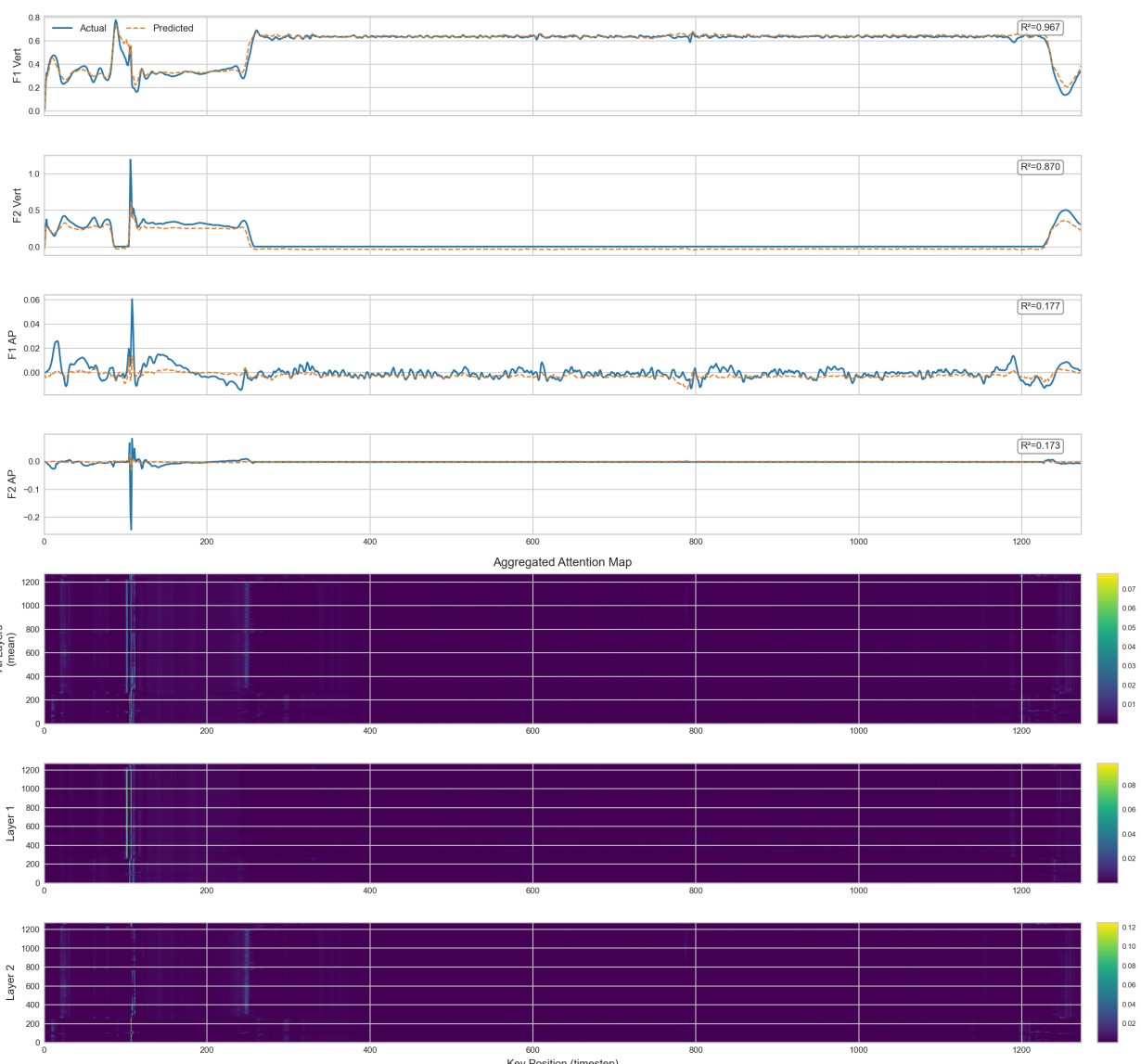

Figure 21: Synchronized prediction and attention for the best GL trial ($R^2 = 0.967$). Top panels: predicted (orange) vs. actual (blue) GRF for four channels. Bottom panels: aggregated attention, Layer 1, and Layer 2 heatmaps (query $\times$ key). Attention mass concentrates at gait events (heel-strike, toe-off).

