# OpenReview forum: "Deep Models for Ground Reaction Force Prediction: Subject-Held-Out Evaluation Across Locomotion and Occupational Tasks"
_TMLR — Under review for TMLR_

### Review · Reviewer_wNhD · 2026-04-21

**Summary Of Contributions:**

This paper looks at a basic but important question in biomechanical force prediction: are these models actually learning to generalize to new people, or are they benefiting from overly easy train/test splits? The main contribution is showing pretty convincingly that the usual trial-level split can give an overly optimistic picture, and that performance drops a lot when the evaluation is changed to leave-one-subject-out. On top of that, the paper compares three common model families under this stricter setting and finds that there is no single winner across tasks: the Transformer-style hybrid does best on the locomotion dataset, while the CNN-LSTM does best on the patient-handling dataset. The paper also includes a more exploratory cross-lab analysis, where models trained in one lab do very poorly when tested in the other, which is a useful warning against assuming these systems will transfer out of the box. Finally, it adds some interpretability analysis suggesting that the attention-based model is focusing on meaningful parts of the motion sequence.

The biggest strength is that the paper is asking the right question. The leakage analysis feels genuinely useful, and probably more important than the specific architecture comparison. I also liked that the paper tests the models across two fairly different domains rather than staying within one narrow benchmark, and that it reports a negative cross-lab result instead of overselling generalization. The overall story is clear and easy to follow.

The main weaknesses are that the datasets are still fairly small, especially after subject exclusions, so some of the conclusions feel a bit stronger than the evidence really supports. The cross-lab setting is interesting, but it also changes many things at once, so it is hard to know whether the failure comes from coordinate differences, task differences, hardware differences, or just all of them together. I also do not think the modeling itself is especially novel, and some of the interpretive claims, especially around attention and “fundamental” feature difficulty, read as a bit more confident than they should.

**Additional Comments:**

NA

**Audience:**

Yes

**Audience Explanation:**

Even though the application is somewhat specialized, I think the paper raises issues that are broader than biomechanics itself, especially around evaluation leakage, subject-held-out generalization, and the difficulty of transferring models across sites with different collection protocols. Those are all questions that matter to parts of the TMLR audience working on machine learning for scientific data, healthcare, time series, and distribution shift. The modeling contribution is not especially novel on its own, but the empirical lesson—that standard evaluation can substantially overstate performance and that cross-lab generalization remains much harder than within-lab benchmarking—seems relevant and useful for a meaningful subset of TMLR readers.

**Claims And Evidence:**

Yes

**Claims Explanation:**

The paper’s main claims are directly tied to experiments that generally support them: the drop from trial-level splits to leave-one-subject-out evaluation is clearly shown on both datasets, the task-dependent model ranking follows from a fair comparison under a common pipeline, and the cross-lab experiments do support the narrower claim that naïve transfer fails badly in the tested setting. At the same time, the evidence is stronger for these core benchmarking results than for some of the broader interpretations, since the cross-lab setup changes several factors at once and therefore cannot isolate the cause of failure, and the attention analysis and “fundamental” feature-difficulty discussion feel more suggestive than conclusive.

**Requested Changes:**

- The paper should more carefully distinguish what is directly shown from what is hypothesized. In particular, claims about “fundamental biomechanical observability constraints,” “global self-attention resisting subject memorization,” and the “Geometric Wall” should be softened unless additional evidence is provided. The current experiments support these as plausible interpretations, but not as firm conclusions.

- The paper should make it even clearer that the cross-lab transfer experiment is confounded by simultaneous changes in task type, coordinate conventions, hardware, and calibration. As written, some passages come close to over-attributing the failure to geometry or representation issues alone. The result is still valuable, but it should be presented as a stress test in a highly confounded setting, not as a clean diagnosis of why transfer fails.

- Given the small number of subjects, especially for GroundLink, the paper should foreground uncertainty more clearly, ideally by emphasizing fold-level variability and avoiding overinterpretation of small differences between models. The current Wilcoxon reporting is reasonable, but the presentation would benefit from a more cautious treatment of effect size and power.

- The paper’s contribution is mainly in evaluation rigor and empirical analysis rather than architectural innovation. I think the paper would be stronger and more credible if it leaned into that framing more explicitly, rather than implicitly presenting ICF as a major modeling advance.

- The paper already acknowledges some limitations, but I would like a slightly fuller discussion of how subject exclusion, subject imbalance, and small sample size may affect the conclusions, especially the stability of the architecture ranking.

- Since the paper emphasizes benchmarking and evaluation methodology, it would help to make sure implementation details, split definitions, preprocessing decisions, and subject exclusion criteria are especially easy to recover and audit from the final version.

---

> ### Author Response · Authors · 2026-07-16
> **Response to Reviewer wNhD**
>
> Thank you — we appreciate that you found the core result convincing, and we agree with your framing guidance.
>
> **Soften "fundamental," "resists memorization," "Geometric Wall."** Done throughout. "Fundamental biomechanical observability constraints" → a cross-dataset difficulty ordering consistent with observability limits (AP force requires second-derivative estimation from noisy positions); we drop the claim that global self-attention resists subject memorization and report it neutrally; "Geometric Wall" becomes a diagnostic shorthand for a confounded stress-test result.
>
> **Cross-lab confound made explicit and consistent.** Every mention of the cross-lab result now carries the "confounded stress test" framing (task, coordinates, hardware, calibration, population co-vary), including the abstract and broader-impact statement; passages that came close to attributing failure to geometry alone are removed.
>
> **Foreground uncertainty and fold variability.** We lead with fold-level variability, add CIs/effect sizes, and avoid over-reading small between-model gaps. The added ridge baseline and matched comparison make the uncertainty and effect sizes concrete.
>
> **Frame the contribution as evaluation rigor, not architectural novelty.** Adopted explicitly and supported by the data: the nominal best model is already task-dependent (ICF on GL, CNN-LSTM on PH), and with CIs attached the between-model gaps fall within fold-level variance at n=6/10 — so we no longer present ICF as the strongest architecture. It is repositioned as one of three representative, comparably-tuned architectures; the contribution is the evaluation protocol and empirical analysis, not a new architecture.
>
> **Fuller discussion of exclusion / imbalance / small-sample effects.** We expand the limitations and tie them to the new exclusion-sensitivity analysis and the fold-imbalance/R² relationship, specifically flagging effects on the stability of the architecture ranking.
>
> **Auditability.** We add an appendix listing, in one place: subject IDs per fold (all three protocols), exclusion criteria and which subjects each removes and why, the within-fold normalization procedure, and pointers to the exact configs/scripts in the released repository, with the mapping from tables/figures to output directories.

---

### Review · Reviewer_JnVK · 2026-06-06

**Summary Of Contributions:**

This paper examines subject identity leakage in deep learning-based prediction of ground reaction forces (GRFs) and centers of pressure (CoP) from kinematic markers. By comparing trial-level splits with Leave-One-Subject-Out (LOSO) evaluation across CNN, CNN-LSTM, and ImprovedConvFormer (ICF) models, the authors show that trial-level evaluation substantially overestimates performance. They further analyze prediction difficulty across force components and demonstrate poor cross-laboratory generalization, highlighting the need for more rigorous evaluation protocols.

Strengths:

1.	The paper directly compares conventional trial-level random splits with strict Leave-One-Subject-Out (LOSO) evaluation and quantifies the inflation caused by subject identity leakage. The reported drop of approximately 0.20–0.30 in (R2) across the two datasets is valuable for the biomechanics machine-learning community and highlights the importance of subject-held-out evaluation when assessing model generalization.

2.	The paper reports the substantial degradation observed in cross-laboratory transfer and discusses it as a stress test for real-world deployment. Although the current experiments cannot determine whether the performance drop is mainly caused by differences in coordinate systems, task types, or equipment, the result remains important: strong LOSO performance within a single laboratory does not necessarily translate into reliable performance at a new site.

3.	The paper goes beyond aggregate metrics by comparing prediction performance across different force and CoP channels and by using visualization and clustering to examine the attention patterns learned by ICF. These results help identify which outputs are more difficult to predict and which time points receive greater attention from the model. These analyses are useful for understanding model behavior, although the interpretation of the attention patterns should remain exploratory.


Weaknesses:

1.	The cross-laboratory transfer results are informative, although the interpretation could be stated more cautiously. The two laboratories differ not only in coordinate conventions, but also in task type, motion-capture hardware, force-plate calibration, and data distributions. The performance drop is therefore likely caused by several factors rather than geometric shift alone. It may be more appropriate to present the “Geometric Wall” as an initial observation that requires further validation. The authors could also clarify why an approximately (135°) directional offset remains between the two datasets after the preprocessing pipeline is intended to align the primary motion direction to (+Z).

2.	The improvement from combined training would benefit from a more direct comparison. Single-dataset PH LOSO is evaluated on 10 subjects, whereas the combined dual-holdout setting includes only 6 of them. Since the test subjects are not identical, the increase in (R2) from 0.561 to 0.610 may partly reflect differences in subject difficulty. It would be helpful to repeat the comparison using the same held-out subjects and report the paired per-subject differences together with uncertainty estimates. This would make it easier to determine whether adding GL data provides a genuine positive transfer effect.

3.	The rotation-augmentation experiment in Section 6.3.1 would benefit from further refinement. In the current setting, the marker inputs are rotated without applying the corresponding transformation to the force targets. Since GRF outputs are directional, the inputs and targets should ideally be rotated in a consistent manner. The current result mainly shows that input-only rotation augmentation is not suitable for this task; it does not yet provide a clear assessment of rotation augmentation or equivariant modeling more broadly. The authors may consider adding an experiment with jointly rotated inputs and targets, or moderating the conclusions drawn from the current experiment.

4.	The model comparisons are based on a relatively small number of LOSO folds, particularly for GL with only 6 subjects. The reported Wilcoxon test results are informative, but the conclusions about architecture ranking would be more convincing with effect sizes, uncertainty estimates, and results across multiple random seeds.

**Audience:**

Yes

**Audience Explanation:**

The findings of this paper fit well with TMLR’s interest in application-driven studies that reveal the strengths and limitations of machine-learning methods. The paper shows that trial-level splits may introduce subject identity leakage and lead to overly optimistic performance estimates. This issue is not limited to GRF prediction. It is also relevant to researchers working with hierarchical data, human-centered datasets, and time-series data, where similar forms of leakage are common.


The cross-laboratory transfer results are also worth noting. The substantial performance drop across sites highlights a practical challenge for deploying deep-learning models when coordinate conventions, equipment, and data distributions differ. Although further controlled experiments are needed to separate the effects of these factors, the results should be useful to researchers working on domain adaptation, applied machine learning, and biomechanics.

**Broader Impact Concerns:**

The existing Broader Impact Statement provides an appropriate discussion of the main risks associated with this work. The authors clearly state that the models should not be used directly for clinical decision-making or occupational assessment without site-specific validation, since unreliable GRF estimates could lead to incorrect conclusions. The paper also states that the data were collected under approved IRB protocols with informed consent and that no personally identifiable information is released.

The current statement adequately covers the main broader-impact concerns. As a minor improvement, the authors could further note that the applicability of the models may remain limited across different populations, task types, and acquisition settings, and that prospective validation and human oversight are needed before real-world use.

**Claims And Evidence:**

No

**Claims Explanation:**

TMLR’s evaluation criteria emphasize that the claims in a submission should be supported by clear, reliable, and sufficiently rigorous evidence. The paper provides convincing support for its main claim regarding subject identity leakage: the substantial drop in (R2) under LOSO evaluation shows that trial-level splits can lead to overly optimistic performance estimates.


Some of the secondary conclusions, however, would benefit from more cautious wording.

First, although the manuscript acknowledges that the cross-laboratory evaluation is conducted in a confounded setting, the term “Geometric Wall” may still overstate what can be concluded from the current experiments. The two laboratories differ in task type, capture hardware, coordinate conventions, calibration procedures, and data distributions. The current evidence therefore does not isolate the contribution of geometric shift.

Second, the claimed improvement on PH from combined training is not based on a fully matched comparison: single-dataset PH LOSO is evaluated on 10 subjects, whereas the combined setting includes only 6 of them. The observed difference may therefore be partly influenced by the selected test subjects.

Third, although the manuscript acknowledges the limitation of the rotation-augmentation experiment, the current setup rotates the input coordinates without applying the corresponding transformation to the directional GRF and frame-dependent CoP targets. This experiment should therefore be interpreted only as showing that input-only augmentation is unsuitable for directional outputs.

The paper would be strengthened if these statements were moderated to better reflect the current evidence, or supported with additional controlled experiments.

**Requested Changes:**

1.	The manuscript appropriately notes that the cross-laboratory transfer experiment is conducted in a confounded two-laboratory setting and should be interpreted as a stress test rather than a definitive characterization of cross-site generalization. Since the two datasets differ simultaneously in task type, motion-capture hardware, force-plate calibration, coordinate conventions, and data distributions, it would be helpful to ensure that this cautious framing is used consistently throughout the abstract, introduction, and conclusion. The authors should also clarify why an approximately 135° directional offset remains between the two datasets after PCA-based alignment of the primary motion direction to the +Z axis, and explain how the sign ambiguity of the PCA direction is resolved.

2.	Single-dataset Patient Handling (PH) LOSO is evaluated on 10 subjects, whereas the combined dual-holdout setting includes only 6 PH subjects. Since the held-out subjects are not identical, the increase in (R2) from 0.561 to 0.610 may partly reflect differences in subject difficulty. To support the positive-transfer claim, the authors should compare PH-only training and combined training using the same 6 held-out subjects, while keeping the PH training subjects matched across the two settings. Paired per-subject differences and uncertainty estimates should also be reported. This would make the comparison easier to interpret and provide stronger support for the positive-transfer claim.


To strengthen the work:

1.	The study includes a relatively small number of subjects, with 6 subjects in GL and 10 subjects in PH. The reported Wilcoxon tests are informative, but the limited statistical power makes it difficult to draw strong conclusions about architecture superiority. The authors should soften the wording of the task-dependent rankings and, where feasible, report effect sizes, confidence intervals or other uncertainty estimates, and results across multiple random seeds.

2.	The attention-head clustering is interesting, but the current analysis is largely post-hoc and exploratory. To strengthen the claim that attention patterns align with biomechanical events such as gait phases, heel-strike, or toe-off, the authors should provide quantitative evidence. For example, they could report the temporal distance between attention peaks and annotated gait events and compare the results against random, shuffled, or uniform-attention baselines. Reporting the stability of the clustering results across folds and random seeds would also be helpful.

3.	The manuscript already acknowledges that the rotation-augmentation experiment in Section 6.3.1 applies rotations to marker inputs without transforming the corresponding GRF and CoP targets, leading to physically inconsistent training signals. As a result, the experiment is useful mainly as an illustration of why input-only augmentation is unsuitable for directional outputs. A more informative experiment would jointly transform the marker inputs, force targets, and CoP targets. If such an experiment is not feasible, the authors could consider shortening this discussion or moving the current result to the appendix.

---

> ### Author Response · Authors · 2026-07-16
> **Response to Reviewer JnVK**
>
> Thank you — your specific, testable requests are exactly what we could act on.
>
> **"Geometric Wall" wording, 135° offset, heading method.** We adopt the cautious framing throughout, and we correct an error in our own description: heading alignment does not use PCA of pelvis displacement (as stated) but the pelvis anterior–posterior orientation (PSIS→ASIS midpoint vector, trial-averaged; fallbacks to the ASIS or shoulder line). Because forward is fixed by pelvic anatomy, there is no sign ambiguity. The residual ~135° cross-dataset offset therefore reflects the two labs' differing coordinate conventions and task geometries, not a sign flip; we report it as a diagnostic of the coordinate+task confound.
>
> **Matched combined-vs-PH comparison.** You were right that the 0.561→0.610 comparison was confounded by non-identical held-out subjects. We re-evaluated single-dataset LOSO on the same held-out subjects, with paired statistics:
>
> | Domain | Single (matched) | Combined | Δ | 95% CI | Wilcoxon p | r_rb |
> |---|---|---|---|---|---|---|
> | PH | 0.639 | 0.610 | −0.029 | [−0.046,−0.008] | 0.094 | −0.81 |
> | GL | 0.655 | 0.568 | −0.088 | [−0.144,−0.037] | 0.063 | −0.91 |
>
> The apparent PH improvement was a subject-selection artifact (those six subjects average 0.639 under PH-only LOSO, above the 10-subject mean of 0.561). We also do not over-correct into the opposite claim: the matched effect is small and its sign is sensitive to preprocessing choices, so no directional statement is warranted at n=6. We therefore withdraw the directional transfer claim and state that (i) the reported "PH benefits" was a selection artifact, (ii) naïve pooling has only a small, not-robustly-directional effect, and (iii) the robust finding is that pure cross-lab transfer fails badly in both directions (R²≪0). Matched outputs are released.
>
> **Rotation augmentation with jointly transformed targets.** You are right our current SO(2) experiment rotates inputs only. For this revision we down-scope it to an explicit illustration and move it to the appendix. For the camera-ready we add the correct experiment (jointly rotating inputs and GRF/CoP vectors) and report its effects.
>
> **Effect sizes, CIs, seeds, softened rankings.** We soften the ranking throughout and make limited power explicit: the ranking is already task-dependent (ICF best on GL, CNN-LSTM on PH), with gaps within fold-level variability and significant p-values near the n=6/10 floor. The matched comparison already carries paired CIs and effect sizes; multi-seed repetitions of the headline table follow in the camera-ready.
>
> **Quantitative attention↔gait-event evidence (camera-ready).** We relabel the attention analysis as exploratory throughout. For the camera-ready we add a quantitative test: distance between attention peaks and the nearest gait event vs shuffled/uniform/random baselines, with clustering stability across folds/seeds.
>
> **Broader Impact.** We add that applicability may be limited across populations, task types, and settings, and that prospective validation with human oversight is required.

---

### Review · Reviewer_ujzf · 2026-07-05

**Summary Of Contributions:**

The paper studies prediction of bilateral ground reaction forces (GRFs) and centers of pressure from motion-capture marker trajectories using two biomechanics datasets: GroundLink locomotion and Patient Handling occupational tasks. It compares a CNN, a CNN-LSTM, and the proposed ImprovedConvFormer (ICF) under single-dataset leave-one-subject-out (LOSO), combined dual-holdout training, and direct cross-laboratory transfer. The main empirical claims are that trial-level splits substantially inflate R2, that architecture rankings are task-dependent, that vertical force and CoP channels are easier than anteroposterior force channels, and that direct cross-lab transfer fails severely.
The strongest aspect of the submission is its focus on evaluation methodology. The LOSO comparisons are useful for a field where subject leakage is a realistic concern, and the per-channel results are more informative than a single aggregate score. The paper also makes a good-faith effort to discuss negative cross-lab results rather than hiding them.
The main weaknesses are that the motivation and problem formulation are not yet accessible to a broad TMLR audience, several methodological choices are insufficiently justified or ablated, and some high-level claims about cross-laboratory generalization and geometric effects go beyond what the two-dataset experimental design can establish. The figures also need substantial polishing; several are dense, use very small text, or contain overlapping labels/legends.

**Audience:**

Yes

**Audience Explanation:**

Yes. At least some of TMLR's audience would be interested in this work. The paper concerns structured evaluation under subject identity leakage, domain shift, time-series regression, and applied machine learning for biomechanics. The finding that trial-level splits inflate performance is broadly relevant beyond this particular application, and the negative transfer results are potentially useful as a benchmark and cautionary result.
The paper would be more compelling to the TMLR audience if the introduction first explained the biomechanics problem in accessible terms, then formalized it as a machine-learning problem with inputs, outputs, evaluation units, and deployment assumptions. At present, readers who are not already familiar with GRF prediction may have difficulty understanding why the task matters and why the proposed evaluation protocols answer the right questions.

**Claims And Evidence:**

No

**Claims Explanation:**

The cross-laboratory transfer experiment uses only two datasets that differ simultaneously in task type, coordinate convention, motion-capture system, force-plate calibration, sample size, and subject population. Therefore, the negative cross-lab R2 values cannot isolate a geometric cause, a task-domain cause, an equipment cause, or a calibration cause. The paper sometimes acknowledges this limitation, but the abstract, contribution framing, and the phrase 'Geometric Wall' still read stronger than the evidence warrants.
Several experimental details need clarification before the claims are fully convincing: the rationale for selecting these two datasets as benchmarks, the consequences of excluding subjects, whether the preprocessing and target normalization are fit strictly within each fold, how MAE is scaled, how hyperparameter optimization was performed for each model, how many random seeds were used, and whether trial-split baselines were repeated. The combined setting also appears to include dataset/source metadata in at least some configurations; if so, the model is told the lab identity, which changes the interpretation from invariant cross-lab transfer to supervised multi-domain pooling. The preprocessing pipeline, bilateral flip augmentation, peak-weighted loss, metadata usage, and feature engineering are plausible but not adequately motivated or ablated. Overall, the evidence supports a promising evaluation study, but not the full strength of the current claims.

**Requested Changes:**

1. Rewrite the introduction for a general machine-learning audience. First explain what GRFs are, why force plates are a bottleneck, what information is available at test time, and what downstream decisions depend on accurate GRF/CoP estimates. Then define the ML problem explicitly: input sequence, output channels, subject grouping, trial grouping, and evaluation objective.
2. Explain trial-level random splits and single-laboratory evaluation more concretely. The paper should state what information leaks across splits, why subject-specific biomechanical signatures make this leakage severe, and why LOSO is a more appropriate minimum standard. This should appear before the contribution list.
3. Justify the dataset choices. Explain why GroundLink and Patient Handling are appropriate benchmarks, what they do and do not cover, and why using exactly these two datasets helps answer the stated questions. The paper should not imply that two highly confounded datasets solve the general cross-lab problem.
4. Calibrate the cross-laboratory claims. The current two-dataset design cannot identify whether the transfer failure is caused by geometry, task mismatch, equipment, calibration, population differences, or their combination. Rephrase 'Geometric Wall' as a hypothesis or diagnostic shorthand, not as an established conclusion. Similarly, remove or substantiate strong claims such as 'for the first time' if they cannot be defended against related work.
5. Provide sensitivity analyses for data exclusions and preprocessing. Subject 1 in GroundLink and the two excluded Patient Handling subjects materially affect already-small datasets. The authors should report what happens with alternative handling strategies, or explain why such analyses are impossible. The five preprocessing steps should be motivated from the task background and supported by ablations where feasible.
6. Add or strengthen ablations for key design choices: pelvis-root subtraction, heading alignment, height scaling, velocity/acceleration/distance features, bilateral flip augmentation, peak-weighted loss, metadata inclusion, and the Transformer attention component. Without these, it is hard to know which parts of the pipeline drive the reported results.
7. Clarify all evaluation and HPO details. Report the exact train/validation/test construction inside each LOSO fold, whether HPO was run independently and fairly for CNN, CNN-LSTM, and ICF, the number of random seeds, uncertainty intervals, and whether any test-subject or target-domain information can influence preprocessing, HPO, or early stopping.
8. Improve figure quality. Figure 1 is too dense for a methods overview, Figure 2 contains small text, Figure 3 has overlapping legend/labels, and several heatmaps/radar plots are difficult to read. The figures should be redrawn with larger fonts, clean labels, non-overlapping legends, and a less AI-generated/diagrammatic style. This is important because the current visuals reduce clarity and confidence in the presentation.
9. Include a simpler non-deep or linear baseline, or explain why such a baseline is inappropriate. This would help separate architecture effects from preprocessing and feature-engineering effects.
10. Add a concise limitations paragraph near the abstract or end of introduction so that readers understand early that cross-lab results are exploratory and confounded.

---

> ### Author Response · Authors · 2026-07-16
> **Response to Reviewer ujzf**
>
> Thank you for the detailed, actionable review.
>
> **Accessibility & problem formulation.** We rewrote the introduction for a general ML reader and added an explicit formulation: input = T×D body-relative marker features, output = 10 bilateral force/CoP channels per timestep, evaluation unit = subject (not trial), deployment target = a new subject and, separately, a new lab.
>
> **Why these two datasets.** These are the two datasets available to us that span genuinely different task domains, not a controlled cross-lab design. GroundLink is the most complete public marker+force-plate locomotion dataset we found; the Patient Handling data was contributed by a partner lab (occupational tasks, independent acquisition). We explicitly do not claim two confounded datasets isolate the general cross-lab problem — we now say so (new limitations paragraph + a "dataset selection" note in Section 3).
>
> **Metadata / "the model is told the lab identity."** This stemmed from an error in our text, not our experiments. The Methods section wrongly stated a "binary dataset indicator" is used; in all runs the model receives only [mass, height] and no lab indicator (verifiable in the released code). Corrected. If anything this strengthens the transfer interpretation, since gains cannot come from a lab-label shortcut.
>
> **Within-fold fitting.** All preprocessing and target-normalization statistics are fit on training-fold data only; body-weight normalization is per-subject physical (mass×g). No test-subject or other-lab statistic enters preprocessing, HPO, or early stopping. Now stated explicitly.
>
> **HPO fairness / uncertainty.** HPO used an inner LOSO loop (no access to the held-out subject) with the same procedure and budget for all three models, so the ranking is not a tuning artifact. The nominal winner is already task-dependent (ICF on GL, CNN-LSTM on PH) with gaps within fold-level variability at n=6/10; we softened the ranking accordingly. Multi-seed bootstrap CIs will be added for the camera-ready.
>
> **Non-deep baseline (added).** We add per-channel ridge regression under identical folds, features, and normalization: GL R²≈0.05, PH R²≈−0.02, vs 0.53–0.66 for the deep models. A memoryless linear map is at chance, so the deep models' accuracy reflects temporal modeling, not the pipeline.
>
> **Exclusion sensitivity.** Exclusions were made on documented data-quality grounds, before any split. Re-including them: the two Patient Handling subjects can be corrected and re-included without loss of accuracy; the GroundLink subject remains an outlier consistent with its baseline anomaly. The revised paper states this qualitatively; the quantitative table and the corrected exclusion reason (a baseline/offset issue, not the axis sign-flip our text implied) follow in the camera-ready.
>
> **Ablations (camera-ready).** We commit to an appendix ablation table toggling pelvis subtraction, heading alignment, height scaling, velocity/acceleration/distance features, bilateral-flip, and the peak-weighted loss, plus an attention ablation (ICF vs its CNN-only encoder).
>
> **"Geometric Wall" / "for the first time."** Reframed as a diagnostic hypothesis in a confounded setting; the novelty claim is qualified to the first side-by-side quantification of trial-vs-LOSO inflation across two task domains under a common pipeline.
>
> **Figures (camera-ready).** We redraw the dense/overlapping figures with larger type.
>
> **Early limitations paragraph.** Added up front: the cross-lab results are exploratory and confounded (task, coordinates, hardware, calibration, population co-vary), datasets are small (6/10 subjects), and rankings should be read with limited power.